# GGGGCC microsatellite RNA is neuritically localized, induces branching defects, and perturbs transport granule function

Alondra Schweizer Burguete[1]*, Sandra Almeida[2], Fen-Biao Gao[2], Robert Kalb[3], Michael R Akins[4], Nancy M Bonini[1]*

[1]Department of Biology, University of Pennsylvania, Philadelphia, United States; [2]Department of Neurology, University of Massachusetts Medical School, Worcester, United States; [3]Division of Neurology, Department of Pediatrics, Children's Hospital of Philadelphia, University of Pennsylvania School of Medicine, Philadelphia, United States; [4]Department of Biology, Drexel University, Philadelphia, United States

**Abstract** Microsatellite expansions are the leading cause of numerous neurodegenerative disorders. Here we demonstrate that GGGGCC and CAG microsatellite repeat RNAs associated with *C9orf72* in amyotrophic lateral sclerosis/frontotemporal dementia and with polyglutamine diseases, respectively, localize to neuritic granules that undergo active transport into distal neuritic segments. In cultured mammalian spinal cord neurons, the presence of neuritic GGGGCC repeat RNA correlates with neuronal branching defects, and the repeat RNA localizes to granules that label with fragile X mental retardation protein (FMRP), a transport granule component. Using a *Drosophila* GGGGCC expansion disease model, we characterize dendritic branching defects that are modulated by FMRP and Orb2. The human orthologs of these modifiers are misregulated in induced pluripotent stem cell-differentiated neurons (iPSNs) from GGGGCC expansion carriers. These data suggest that expanded repeat RNAs interact with the messenger RNA transport and translation machinery, causing transport granule dysfunction. This could be a novel mechanism contributing to the neuronal defects associated with *C9orf72* and other microsatellite expansion diseases.

**\*For correspondence:**
alondraschweizer@gmail.com
(ASB); nbonini@sas.upenn.edu
(NMB)

**Competing interests:** The authors declare that no competing interests exist.

## Introduction

Expansions of short tandem nucleotide repeat sequences termed 'microsatellite repeats' cause various devastating dominantly inherited neurodegenerative disorders, including spinocerebellar ataxias, Huntington's disease, and the myotonic muscular dystrophies (expansion of CAG, and CUG and CCUG repeats, respectively (*Orr and Zoghbi, 2007*)). Most recently, the GGGGCC repeat expansion in the *C9orf72* gene has been shown to be associated with amyotrophic lateral sclerosis/frontotemporal dementia (ALS/FTD) (*DeJesus-Hernandez et al., 2011*; *Renton et al., 2011*). How microsatellite repeat expansions occurring both within coding and non-coding segments of the affected genes cause neuronal degeneration remains a central question in the field.

Microsatellite repeat RNAs are thought to induce neurodegeneration through multiple distinct mechanisms (*Narayan et al., 2014*; *Nelson et al., 2013*). These include both loss and gain of function in the encoded protein (*Blum et al., 2013*); however, a number of disease-associated expanded microsatellite repeats, like (GGGGCC)$_n$, occur in non-coding sequence, suggesting that the RNA product may be toxic (*Belzil et al., 2012*). Nuclear toxicity has been proposed to be a disease

**eLife digest** Genes contain instructions to build proteins, but these instructions are often interrupted by stretches of DNA that do not code for protein. Typically the entire length of a gene is copied to produce an intermediate molecule of RNA, which is processed to remove the non-coding regions before being translated to make a protein. The genes associated with various neurodegenerative diseases, including Huntington's disease and myotonic muscular dystrophies, often also carry short stretches of DNA sequence that are repeated one after the other.

An increase in the number of the repeats within one of these genes can lead to a neurodegenerative disease. These disorders often have similar features, but are associated with different repeat sequences that can occur either in regions of the gene that code for protein or regions that do not. When the repeats lie in a non-coding region, it is thought that the RNA itself and not the protein causes the damage to nerve cells. While it is not known how this happens, it could be related to the shape of the RNA molecules, which in turn controls where the RNA molecules will go within a cell.

Inside nerve cells, some RNAs (but not all) are directed to particles called 'transport granules'. These particles carry specific RNAs into the tips of the nerve fibers where they are then translated into proteins. Burguete et al. wanted to test whether disease-associated RNAs that contain repeats might interfere with this process in nerve cells. Microscopy showed that many RNAs with expanded stretches of repeats ended up in the transport granules by mistake, and were carried toward the tips of the nerve branches.

When the repeat-containing RNAs localized to the transport granules, the nerve endings tended to form fewer branches. By analyzing the components of the granules, Burguete et al. could show that the incorrect localization of RNA molecules in the granules appeared to interfere with the production of proteins in the nerve branches. This disruption could contribute to the nerve cell defects seen in the many neurodegenerative diseases associated with these types of repeat expansions. These data suggest that preventing the disruption of transport granules' activity could represent a potential therapeutic avenue against these diseases.

mechanism mediated either by expanded repeat RNA present in nuclear foci, or by expanded repeat RNA-encoded *r*epeat *a*ssociated *n*on-ATG (RAN) translated peptides (*Haeusler et al., 2014*; *Kwon et al., 2014*; *Zu et al., 2011*; *Jovicic et al., 2015*; *Zhang et al., 2015*; *Freibaum et al., 2015*). However, the RNAs generated from these loci commonly have high structural context (*Napierala and Krzyzosiak, 1997*; *Sobczak et al., 2003*; *Michlewski and Krzyzosiak, 2004*; *Fratta et al., 2012*; *Reddy et al., 2013*), which is a striking feature of cis-acting localization signals that target messenger RNAs (mRNAs) to specific subcellular sites where they can then undergo local translation (*Hamilton and Davis, 2007*; *Martin and Ephrussi, 2009*; *Holt and Schuman, 2013*). Therefore, we hypothesized that such disease-associated RNAs might interact with the mRNA localization and/or translation machinery with deleterious consequences.

Here we show that expanded microsatellite repeat RNAs, including the GGGGCC repeat RNA associated with ALS/FTD, become localized to granules in neurites of mammalian neurons in culture. Such neuritic GGGGCC RNA-positive granules are also present in iPSNs from GGGGCC expansion carriers. This subcellular localization is shared among many expanded repeat RNAs associated with human disease that bear high structural content, including CAG, CUG, and CCUG repeat RNAs. We further show by detailed analysis that at least two of these RNAs—GGGGCC and CAG—become localized to dynamic RNA-granules in neurites. Detailed focus on the GGGGCC repeat RNA revealed neuritic branching defects and suggests the expanded microsatellite repeat RNA may interfere with transport granule function. These data indicate that this property may contribute to the degenerative effects conferred by expanded GGGGCC RNA and additional expanded microsatellite repeat RNAs associated with a wide class of human neurological disorders.

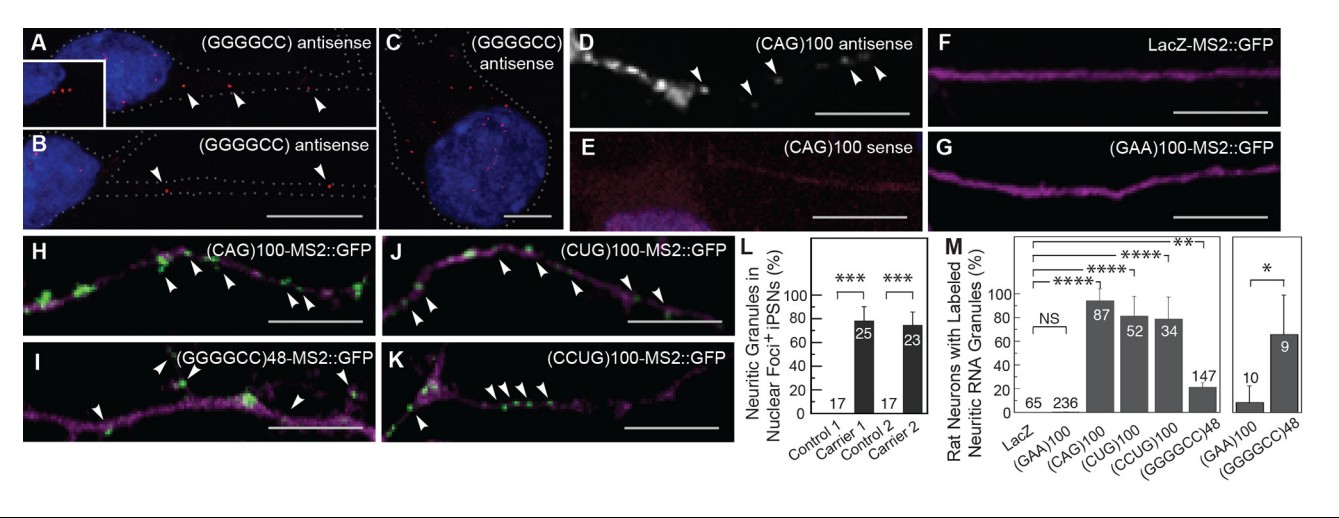

**Figure 1.** The GGGGCC repeat and other microsatellite RNA repeats with high secondary structure content are neuritically localized. (A–B) GGGGCC repeat RNA is neuritically localized in discrete granules in human iPSNs derived from *C9orf72* GGGGCC repeat expansion carriers. (A–C) GGGGCC repeat RNA (red) was detected with a $(GGCCCC)_4$ antisense probe. Neuritic GGGGCC RNA granules (arrowheads) were observed (A) proximally, (inset in A) as linear arrays, and (B) distally. (C) GGGGCC RNA granules in the cell body. (A–C) iPSNs from carrier 2 are shown. Neurites and cell body are outlined (dotted line). (D–E) CAG repeat RNA is localized to neuritic granules in primary rat spinal cord culture. *In situ* hybridization of primary rat spinal cord neurons transfected with $(CAG)_{100}$ RNA. The $(CAG)_{100}$ RNA construct was not MS2-tagged (see *Figure 1—figure supplement 2A* and Materials and methods). Neuritic RNA granules (white; arrowheads) were detected with a (D) $(CUG)_8$ antisense but not with a (E) $(CAG)_8$ sense probe. Distributions shown are representative of two biological replicates. (F–K) Primary rat spinal cord neurons transfected with NLS-CP-GFP (green; arrowheads) and (F) LacZ-MS2, (G) $(GAA)_{100}$-MS2, (H) $(CAG)_{100}$-MS2, (I) $(GGGGCC)_{48}$-MS2, (J) $(CUG)_{100}$-MS2, or (K) $(CCUG)_{100}$-MS2 (see *Figure 1—figure supplement 2A* and Materials and methods for construct details). Whereas (F) the control RNA LacZ-MS2, or (G) an expanded repeat RNA without secondary structure, $(GAA)_{100}$-MS2, did not show GFP accumulations, (H–K) the other expanded repeat RNAs conferred punctate GFP staining indicative of RNA enrichment in neuritic RNA granules. See *Figure 1—figure supplement 1E* for the pattern of expression of the MS2 alone control, which lacked neuritic puncta. DsRed (magenta) was coexpressed to outline the neurons. (L) Quantitation of iPSNs with neuritic GGGGCC repeat RNA granules. Nuclear foci positive (+) neurons were defined as having ≥5 (carrier 2) or ≥1 (carrier 1) nuclear GGGGCC RNA foci. ANOVA *p*-value = 0.00033. (M) Primary rat spinal cord neurons were transfected as in (F–K) and the percentage of neurons with neuritic repeat RNA particles was determined. At right, neurons within the mixed culture with the large morphology characteristic of motor neurons were scored for neuritic particles. ANOVA *p*-value = $1.1 \times 10^{-7}$. (L–M) Numbers indicate the total number of neurons scored from a minimum of (L) two or (M) three biological replicates. Averages ± standard deviation are given. See Materials and methods for statistical analysis. Post-hoc: ****$p < 2.5 \times 10^{-6}$, ***$p < 0.00042$, **$p < 0.0085$, *$p < 0.023$. (D) A single confocal section or (A–C, E–K) Z-series projections. Bars: 10 μm. DAPI: blue. See also *Figure 1—figure supplement 1–2*. ANOVA, analysis of variance; DAPI, 4′,6-diamidino-2-phenylindole; CP-GFP, MS2 RNA-binding coat protein fused with green fluorescent protein; NLS, nuclear localization signal.

The following figure supplements are available for figure 1:

**Figure supplement 1.** Neuritic localization of $(CAG)_{100}$ RNA by *in situ* hybridization.

**Figure supplement 2.** Microsatellite repeat and control expression constructs.

## Results

### Microsatellite repeat RNAs localize to neuritic granules

To explore the idea that expanded repeat RNAs may be localized in neurons, we initially focused on the expanded GGGGCC repeat associated with ALS/FTD (*DeJesus-Hernandez et al., 2011*; *Renton et al., 2011*). The GGGGCC RNA repeat is highly structured, assuming both G-quadruplex and stem-loop conformations (*Fratta et al., 2012*; *Reddy et al., 2013*). We analyzed its localization in iPSNs derived from two *C9orf72* hexanucleotide expansion carriers (carrier 1, line #5; carrier 2, line #11 (*Almeida et al., 2013*)). These neurons contain RNase sensitive nuclear GGGGCC foci specifically in carrier samples, and not in control-derived samples (*Almeida et al., 2013*). We confirmed that iPSNs contained nuclear GGGGCC RNA foci, but also found that 78 ± 12% SD (carrier 1; n=25 neurons) and 75 ± 11% SD (carrier 2; n=23) of iPSNs that contained nuclear GGGGCC RNA foci also

contained neuritic GGGGCC RNA particles by *in situ* hybridization (*Figure 1A,L*). The GGGGCC RNA particles were detected both proximally and distally at over 45 μm from the cell body in neurites and were, in some cases, lined up, consistent with possible association with a cytoskeletal track (*Figure 1A–B*). In addition, GGGGCC repeat RNA particles were detected in the cell body in nearly all iPSNs that also contained GGGGCC RNA nuclear foci (*Figure 1C*, also *Almeida et al., 2013*). We did not detect GGGGCC RNA in control iPSNs, indicating that the non-expanded repeat is either present below the detection level or not stably expressed in wild-type iPSNs. These data thus suggested that endogenous expanded GGGGCC microsatellite repeat RNA was localized to particles in neurites, in addition to localization elsewhere in the cell.

To see if the finding of neuritic localization was a shared property of microsatellite repeats with high secondary structure, we examined a CAG repeat RNA. We expressed an RNA consisting of 100 repeats of the CAG trinucleotide, $(CAG)_{100}$, in primary rat stage E14 mixed spinal cord neurons (*Mojsilovic-Petrovic et al., 2006*) and probed the RNA localization by *in situ* hybridization. Expanded CAG repeat RNA assumes stem-loop secondary structure (*Michlewski and Krzyzosiak, 2004*), confers neurodegeneration (*Li et al., 2008*), and has been noted to assemble into nuclear foci (*Ho et al., 2005*; *Li et al., 2008*; *Wojciechowska and Krzyzosiak, 2011*). As with the GGGGCC microsatellite repeat, we detected the expanded CAG repeat RNA in discrete particles in neurites, and in the cell body (*Figure 1D–E*, *Figure 1—figure supplement 1A–B*). We also noted nuclear foci as previously described (*Figure 1—figure supplement 1A–D*). These data indicated that two distinct expanded microsatellite repeat RNAs—a GGGGCC repeat and a CAG repeat—both highly structured, are incorporated into particles in neurites. Localization of these RNAs to neurites has not been noted previously.

## Neuritic subcellular localization is a common property of highly structured microsatellite repeat RNAs

To further assess whether this subcellular localization to RNA particles in neurites may be a property common to highly structured expanded microsatellite repeat RNAs, we utilized the bipartite MS2 system (*Bertrand et al., 1998*) to examine additional repeat RNAs, as well as a series of control RNAs. We tagged the RNAs with 12 MS2 stem-loops that are recognized by coat-binding protein, which is fused to a nuclear localization signal and to green fluorescent protein (NLS-CP-GFP). When expressed alone, the NLS-CP-GFP signal was predominantly nuclear (not shown). Co-expression of NLS-CP-GFP with either of two control RNAs, the MS2 (*Figure 1—figure supplement 1E*) or LacZ-MS2 (*Figure 1F*, *Figure 1—figure supplement 1F*) RNAs, produced results similar to NLS-CP-GFP alone: there was no enrichment of signal in cellular processes (see *Figure 1—figure supplement 2A* for construct details). We then examined the localization of an expanded repeat RNA that does not assume stem-loop secondary structure, the GAA repeat associated with Friedreich's ataxia (*Sobczak et al., 2003*). $(GAA)_{100}$-MS2 did not alter the distribution of the GFP reporter, indicating this repeat RNA without secondary structure was not localized to neurites (*Figure 1G*, *Figure 1—figure supplement 1G*). RNA particles were neuritic in <0.5% of neurons in cultures expressing NLS-CP-GFP with control LacZ-MS2 or $(GAA)_{100}$-MS2 RNA (*Figure 1M*). In contrast, neurons transfected with NLS-CP-GFP and $(CAG)_{100}$-MS2 had an RNA distribution like that of $(CAG)_{100}$ by *in situ* hybridization (compare *Figure 1H* with 1D, and *Figure 1—figure supplement 1H* with *Figure 1—figure supplement 1A*), with 94.4 ± 9.6% SD (n=3 cultures, 87 neurons total) of cotransfected neurons containing particles in neurites (*Figure 1M*).

Next, we examined $(GGGGCC)_{48}$-MS2 RNA and found it neuritically localized in 21.1 ± 3.7% SD (n=4 cultures, 147 neurons total) of all neuron types in the cultures (*Figures 1I,M*, *Figure 1—figure supplement 1I*), and in 66.6 ± 33.3% SD (n=3 cultures, 9 neurons total) of large neurons with a morphology characteristic of motor neurons (*Figure 1M*, at right; also *Figure 5—figure supplement 1A*). The RNA repeat expansions associated with myotonic dystrophy types I and II—CUG and CCUG, respectively—are also highly structured RNAs that assume stem-loop conformation (*Napierala and Krzyzosiak, 1997*; *Sobczak et al., 2003*). Indeed, $(CUG)_{100}$-MS2 and $(CCUG)_{100}$-MS2 RNA particles were also present in neurites in over 75% of the transfected neurons (*Figure 1J-K,M*, see also *Figure 1—figure supplement 1J–K*). Thus, in contrast to the control RNAs (MS2, LacZ-MS2, and $(GAA)_{100}$-MS2), multiple microsatellite RNA repeats (CAG, GGGGCC, CUG and CCUG) with high structural context became localized to RNA particles in neurites, by independent detection methods and in a variety of neural systems.

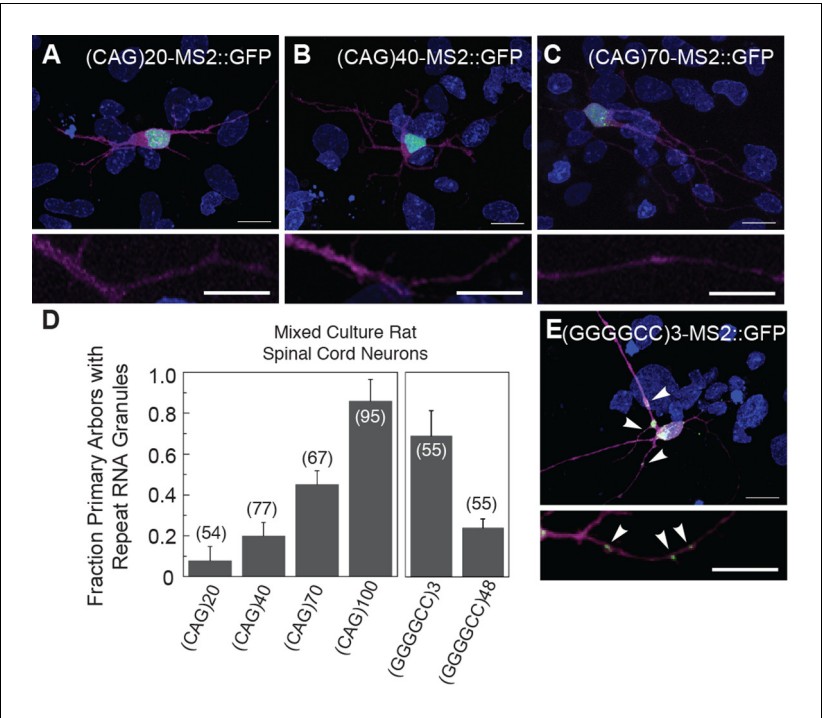

**Figure 2.** Particle formation dependence on CAG and GGGGCC RNA repeat number. (**A–C, E**) Rat mixed spinal cord neurons were transfected with NLS-CP-GFP (green; arrowheads) and (**A**) $(CAG)_{20}$-MS2, (**B**) $(CAG)_{40}$-MS2, (**C**) $(CAG)_{70}$-MS2, or (**E**) $(GGGGCC)_3$-MS2. $(CAG)_{100}$-MS2 and $(GGGGCC)_{48}$-MS2 were transfected as above and are shown in *Figure 1H* and *Figure 1—figure supplement 1H*, and in *Figure 1I* and *Figure 1—figure supplement 1I*, respectively. See *Figure 1—figure supplement 2A* and Materials and methods for construct details. (**D**) Quantitation of the fraction of primary arbors containing ≥1 GFP granule. Parentheses indicate the total number of primary branches counted in three biological replicates. A minimum of 15 transfected neurons were scored for neuritic GFP granules in total. Averages ± standard deviation are given. Data are representative of a minimum of three biological replicates. Confocal Z-series projections. DsRed (magenta) was coexpressed to outline the neurons. Bottom panels: High magnification of neuronal processes. Bars: top panels, 15 µm; bottom panels,10 µm. DAPI: blue. DAPI, 4',6-diamidino-2-phenylindole; CP-GFP, MS2 RNA-binding coat protein fused with green fluorescent protein; NLS, nuclear localization signal.

Disease severity and age of onset in patients with trinucleotide repeat expansion disorders (e.g. CAG and CUG) correlates with increasing repeat number (*Orr and Zoghbi, 2007*). Therefore, we examined the dependence of particle formation on repeat number for the MS2-tagged CAG and GGGGCC RNAs in mixed rat spinal cord neurons, focusing on neurons that contained at least one neuritic RNA particle. The fraction of primary arbors that had particles containing RNAs of 20, 40, 70, and 100 CAG repeats were 0.08 ± 0.07 SD, 0.20 ± 0.06 SD, 0.45 ± 0.07 SD, and 0.86 ± 0.10 SD, respectively (*Figure 2A–D*). These data indicate that, for CAG repeat RNA, there is repeat length specificity for neuritic localization as the prevalence of neuritic particles was highly correlated with increasing repeat number. In contrast, the fraction of primary arbors with $(GGGGCC)_3$-MS2 RNA particles (*Figure 2E*) was higher (0.69 ± 0.12 SD) than the fraction with $(GGGGCC)_{48}$-MS2 particles (0.24 ± 0.04 SD) (*Figure 2D*), and the percentage of neurons in the mixed culture with neuritic particles was also higher for $(GGGGCC)_3$-MS2 (46.4 ± 22.7% SD [n=3 cultures, 110 neurons total]), than for $(GGGGCC)_{48}$-MS2 (21.1 ± 3.7% s.d. (n=4 cultures, 147 neurons total). These data indicate that three GGGGCC units, at the low end of non-expanded *C9orf72* alleles (*DeJesus-Hernandez et al., 2011*; *Gijselinck et al., 2012*; *Renton et al., 2011*; *van der Zee et al., 2013*), are sufficient to confer neuritic localization, and that targeting information is retained in expanded GGGGCC repeat RNA. These data may also suggest that expanded GGGGCC repeat RNA is less efficiently incorporated into RNA granules, or that arbors with expanded GGGGCC repeat RNA had degenerated (hence a lower fraction of arbors with particles).

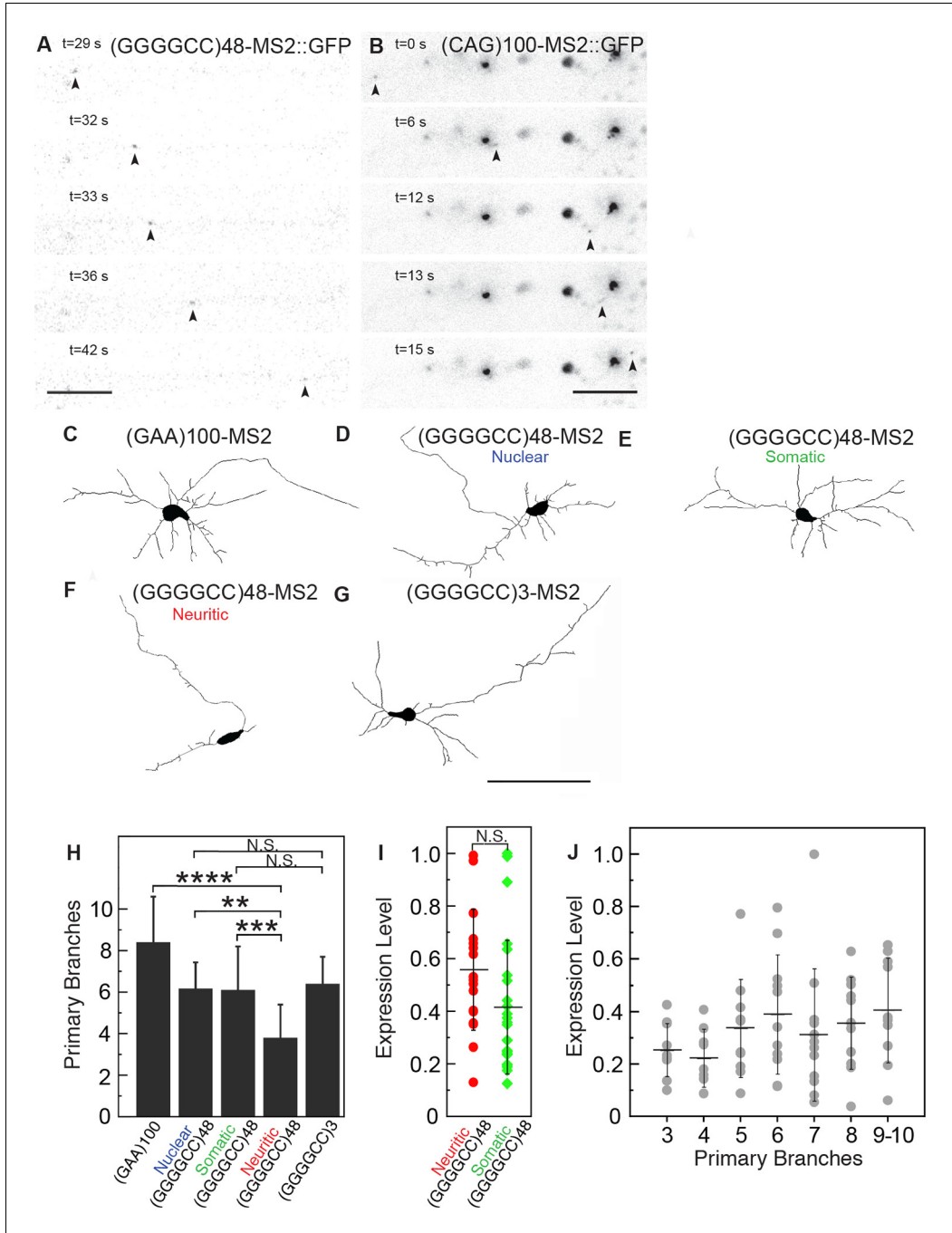

**Figure 3.** (GGGGCC)$_{48}$ and (CAG)$_{100}$ RNA assembles into neuritic transport particles and neuritic (GGGGCC)$_{48}$ RNA correlates with branching defects. (**A–B**) Rat spinal cord neurons were transfected with NLS-CP-GFP and (**A**) (GGGGCC)$_{48}$-MS2 or (**B**) (CAG)$_{100}$-MS2, and the trajectory of motile RNA particles along neuronal processes was captured by time-lapse microscopy. The location of an RNA particle (arrowheads) at indicated time points is shown in individual frames. (**A**) The uninterrupted, unidirectional anterograde particle run originates 22 µm from the cell body (which is outside of the shown frames), and ends 18 µm further away (see *Video 1*). (**B**) The uninterrupted, unidirectional anterograde particle run originates 67 µm from the cell body (which is to the left and outside of the shown frames), and ends 44 µm further away (see *Videos 2* and *3*). Larger stationary particles are also seen. (**C–G**) Cultured primary rat spinal cord neurons with neuritically localized (GGGGCC)$_{48}$-MS2 RNA have fewer primary branches. (**C–G**) Tracings depicting the cell body and primary branches of rat mixed spinal cord neurons expressing either NLS-CP-GFP and (**C**) (GAA)$_{100}$-MS2, (**E-F**) (GGGGCC)$_{48}$-MS2, (**G**) (GGGGCC)$_{3}$-MS2, or (**D**) NES-CP-GFP and (GGGGCC)$_{48}$-MS2 RNA (see *Figure 1—figure supplement 2A* and Materials and methods

*Figure 3 continued on next page*

*Figure 3 continued*

for construct details). (H) Neurons were transfected as in C–G and the number of primary branches were scored in neurons that had (D) nuclear RNA particles, (E) somatic (but not neuritic) RNA particles, or those that had (F) neuritic RNA particles, as indicated. Repeat construct and GGGGCC repeat RNA localization affected branch number ($p < 0.0001$, ANOVA). For individual comparisons by post-hoc Tukey's multiple comparisons test: ****$p < 0.0001$; ***$p < 0.001$; **$p < 0.01$; N.S. $p > 0.05$. (I) Neurons with neuritic (GGGGCC)$_{48}$-MS2 RNA did not have significantly higher expression level, as determined by ImageJ measurement of mean fluorescence intensity of the neural cell bodies (see Materials and methods), than neurons with somatic (GGGGCC)$_{48}$-MS2 RNA. N.S. $p > 0.07$. (J) Expression level did not affect branch number ($p = 0.2331$, ANOVA). Individual comparisons did not reach significance ($p$ value range: 0.2865 to > 0.9999). (H, J) Only neurons with cell bodies >20 μm and with >2 primary branches were included. Standard deviations are given. Bars: A, 5 μm; B, 10 μm; G, 100 μm. ANOVA, analysis of variance; CP-GFP, MS2 RNA-binding coat protein fused with green fluorescent protein; NES, nuclear export signal; NLS, nuclear localization signal.

The following figure supplement is available for figure 3:

**Figure supplement 1.** Nuclear (GGGGCC)$_{48}$-MS2 foci in transfected rat primary spinal cord neurons.

## GGGGCC and CAG repeat RNAs undergo active neuritic transport

The microsatellite repeat RNAs were present in particles not only close to the neural cell body, but also distally in neuronal processes. This raised the possibility that the RNA-particles in the neurites were being actively transported along the length of the projections. To examine this in detail, we explored the dynamics of the localized RNA particles by performing time-lapse imaging. This approach showed that both (GGGGCC)$_{48}$-MS2 and (CAG)$_{100}$-MS2 particles were undergoing antero-grade, retrograde, and bidirectional movement, both proximally as well as distally along neurites (*Figure 3A–B*, *Videos 1–3*). Similar to previous reports for transport ribonucleoprotein particles (RNPs) and consistent with velocities for dynein or kinesin-mediated transport (*Kiebler and Bassell, 2006*), the mean average velocity of uninterrupted unidirectional movement was 1.06 μm/s for (GGGGCC)$_{48}$-MS2, and 1.30 μm/s for (CAG)$_{100}$-MS2, and the average max velocity was 1.40 μm/s for (GGGGCC)$_{48}$-MS2, and 1.85 μm/s for (CAG)$_{100}$-MS2 (*Table 1*). By contrast, particles that underwent corralled movements had a mean average basal velocity of 0.12 μm/s (*Table 1*). We could not detect motile particles above background in neurons expressing control RNAs LacZ-MS2 or MS2. These data indicate that the microsatellite repeat RNAs could be assembling into mRNA transport granules that are dynamic along the neuronal projections.

## Expanded GGGGCC microsatellite repeat RNA causes neuritic branching defects

The presence of the GGGGCC repeat RNA in distal neuritic particles in *C9orf72* hexanucleotide expansion carrier-derived neurons and in transfected rat spinal cord neurons raised the possibility that

**Table 1.** Behavior of repeat RNA particles in rat spinal cord neurons.

| | Mean average velocity (μm/s) | Average max velocity (μm/s) | Velocity range (μm/s) | Particles tracked in neurites | Particles tracked in cell body | Basal velocity* (μm/s) |
|---|---|---|---|---|---|---|
| (GGGGCC)$_{48}$-MS2::GFP n = 11 cells | 1.06 | 1.40 | 0.32–2.67 | 10 | 15 | 0.11 |
| (CAG)$_{100}$-MS2::GFP n = 2 cells | 1.30 | 1.85 | 0.30–4.73 | 5 | 28 | 0.13 |

Uninterrupted unidirectional anterograde and retrograde particle runs with an average run distance of 5.3 $\mu m$ ((GGGGCC)$_{48}$-MS2), and 6.7 $\mu m$ ((CAG)$_{100}$-MS2), were analyzed. (*)The basal velocity is given as a mean average and was estimated by analyzing five particles that underwent corralled movements with an average net displacement of <0.51 $\mu m$ within 20s. Data are from four (GGGGCC)$_{48}$-MS2 and two (CAG)$_{100}$-MS2 independent live imaging sessions. (GGGGCC)$_{48}$-MS2 and (CAG)$_{100}$-MS2 were co-expressed with NLS-CP-GFP. CP-GFP, MS2 RNA-binding coat protein fused with green fluorescent protein; NLS, nuclear localization signal.

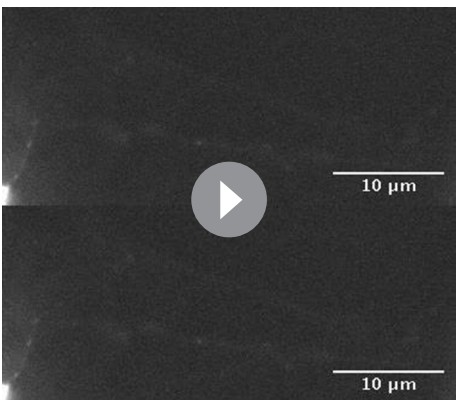

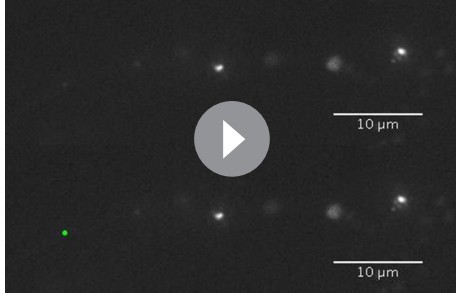

**Video 2.** Movement of a distal (CAG)$_{100}$-MS2 particle. Two identical videos (40 s real-time duration each) are combined vertically; the bottom video displays the tracked particle and its path in green, starting at 67.0 μm and reaching 111.4 μm from the cell body. Images were acquired at 1 frame/s and the video displays at 8 frames/s. The complete caption was 133 s. Selected images are shown in *Figure 3B*.

**Video 1.** Movement of a distal (GGGGCC)$_{48}$-MS2 particle. Two identical videos (60 s real-time duration each) are combined vertically; the bottom video displays a tracked particle in green, starting at 22 μm and reaching 40 μm from the cell body. Images were acquired at 1 frame/s and the video displays at 8 frames/s. The complete caption was 60 s. Selected images are shown in *Figure 3A*.

that such expanded repeat RNA may confer local toxicity. We therefore analyzed neuritic arborization patterns in rat mixed spinal cord neurons with (GGGGCC)$_{48}$-MS2 RNA localized to the nucleus, the soma, or the neurites. While 37.5 ± 3.7% SD (n=108 neurons total) of neurons in the total population contained RNA in the soma (*Figure 1—figure supplement 1I*), about half of these neurons also had neuritically localized (GGGGCC)$_{48}$-MS2 RNA (19.4 ± 7.5% SD of the total neuron population; n=108 neurons total, see also *Figure 1M*), and 10.0 ± 4.0% SD of the total neuron population (n=108 neurons total) contained nuclear (GGGGCC)$_{48}$-MS2 RNA foci (*Figure 3—figure supplement 1A*). Neurons with neuritically localized (GGGGCC)$_{48}$-MS2 RNA had, on average, fewer primary branches (3.8 ± 1.6 SD; n=12 neurons) than neurons with nuclear (GGGGCC)$_{48}$-MS2 RNA foci (6.2 ± 1.3 SD; n=12 neurons), or than neurons with somatic but without neuritic (GGGGCC)$_{48}$-MS2 RNA (6.1 ± 2.1 SD; n=20 neurons) (compare *Figure 3F* with *3D* and *E*, see *Figure 3H*). They also had fewer primary branches than neurons expressing the (GAA)$_{100}$-MS2 control (8.4 ± 2.2 SD; n=21 neurons), which lacked neuritic RNA particles (compare *Figure 3F* with *3C*, see *Figure 3H*). In contrast, neurons with neuritically localized non-expanded (GGGGCC)$_{3}$-MS2 RNA did not show a dramatic primary branch loss *Figure 3G*). These neurons had a similar average number of primary branches (6.4 ± 1.3 SD; n=20 neurons) compared with neurons with nuclear (GGGGCC)$_{48}$-MS2 RNA foci, or neurons with somatic but not neuritic (GGGGCC)$_{48}$-MS2 RNA (*Figure 3H*, compare *Figure 3G* with *3D* and *E*). We did not find a significant correlation between the presence of neuritic (GGGGCC)$_{48}$-MS2 RNA particles and expression level of the RNA in the soma (*Figure 3I*, average normalized mean intensity 0.56 ± 0.23 SD and 0.42 ± 0.25 SD; n=18 and 24 neurons total with neuritic or somatic (GGGGCC)$_{48}$-MS2 RNA, respectively). Similarly, the expression level of the RNA in the soma did

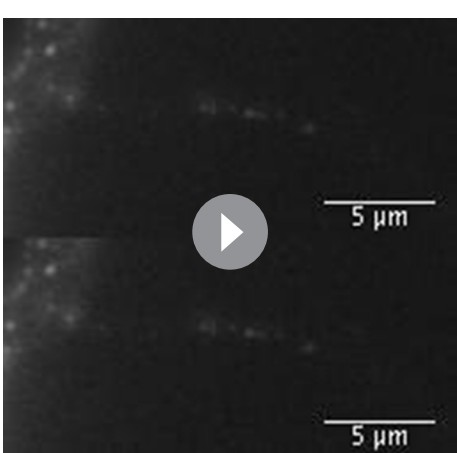

**Video 3.** Movement of a proximal (CAG)$_{100}$-MS2 particle. Two identical videos (97 s real-time duration each) are combined vertically; the bottom video displays the tracked particle and its path in green, starting in the cell body and reaching 6.5 μm into a neurite. Images were acquired at 1 frame/s and the video displays at 8 frames/s. The complete caption was 221 s.

not affect primary branch number (*Figure 3J*; n = 76 neurons total). These data show that when the (GGGGCC)$_{48}$-MS2 repeat RNA is present in neuritic particles, there is a dramatic reduction in primary neural branches—this is not the case when the RNA is nuclear or somatic. Furthermore, high expression is not required to drive (GGGGCC)$_{48}$-MS2 RNA association into neuritic particles or to induce branching defects. These data argue that the presence of the (GGGGCC)$_{48}$-MS2 RNA in neuritic particles is associated with deleterious effects on neuronal branching.

Expanded repeat RNAs have also been reported to undergo translation into peptide repeat proteins. We looked for RAN translated peptide repeat proteins derived from the repeat RNAs (*Ash et al., 2013*; *Mori et al., 2013*; *Zu et al., 2011*) by immunostain utilizing FLAG, HA and Myc tags encoded 3' of the repeat in the three different reading frames, but were unable to provide evidence for the presence of RAN peptides with these constructs in this system (see *Figure 1—figure supplement 2A* and Materials and methods for construct details). Because we observe a dramatic reduction of primary branches only in neurons with neuritic RNA granules, branching defects in our system may be mediated by neuritically localized expanded repeat RNA. Moreover, our data suggest that the toxicity conferred by the expanded GGGGCC repeat RNA is not simply due to neuritic granule association, given that neuritically localized non-expanded (GGGGCC)$_3$-MS2 RNA did not confer dramatic primary branch loss.

To further address the functional impact of the expanded microsatellite repeat on neuron morphology, we analyzed repeat RNA-induced dendritic degeneration in *Drosophila*. To visualize this, we used fly lines expressing (GGGGCC)$_{48}$ repeat RNA or DsRed control RNA (*Figure 1—figure supplement 2B*) in the highly branched class IV epidermal sensory dendritic arborization (da) neurons (*Grueber et al., 2002*, *2003*). These neurons have a characteristic and elaborate dendritic branching pattern, allowing detailed analysis of branch complexity (*Figure 4A*). Expression of *UAS*-(GGGGCC)$_{48}$ resulted in dramatic dendritic branching defects compared with the *UAS*-DsRed control at late third larval instar (compare *Figure 4B to 4A*, see *Figure 4—figure supplement 1A–B*). To determine whether the defects resulted from compromised growth, degeneration of pre-established dendrites, or both, we scored da neuron morphology at two developmental time points: early and late third larval instar. As the animal body size increases during this time, the dendritic field undergoes expansion (*Figure 4I*; also compare *Figure 4C to 4A*); however, similar total number of intersections, branch segments per order, and number of endings indicated no overall major branch loss (*Figure 4—figure supplement 1C–D*). At early third instar, neurons expressing *UAS*-(GGGGCC)$_{48}$ RNA appeared nearly normal (compare *Figure 4D with 4C*), with a dendrite intersection distribution similar to *UAS*-DsRed control neurons (compare yellow bars in *Figure 4J and 4I*, *Figure 4—figure supplement 1A*). By the late stage, however, dendrites in animals expressing *UAS*-(GGGGCC)$_{48}$ RNA had failed to extend far from the cell body (compare pink bars in boxed areas in *Figure 4I–J*, *Figure 4—figure supplement 1B*), and there was a 42% decrease of distal intersections (140–360 μm from cell body) compared to early stage neurons expressing *UAS*-(GGGGCC)$_{48}$ RNA, coinciding with a 53% loss of higher order branches (orders 13–24) (*Figure 4—figure supplement 1E*). These data indicated that neurons expressing *UAS*-(GGGGCC)$_{48}$ RNA were capable of establishing a complex dendritic arbor; however, they subsequently failed to extend and underwent late-stage degeneration of pre-established branches.

## Transport granule components modulate GGGGCC-induced branching defects in *Drosophila* da neurons

Transport mRNP function is critical for neural health and morphology (*Kiebler and Bassell, 2006*; *Holt and Schuman, 2013*). Our data suggested that the incorporation of the expanded microsatellite repeat into RNA-granules was conferring morphological abnormalities. We first asked whether the GGGGCC expansion might lead to dysregulated expression of RNA binding proteins (*Gerstberger et al., 2014*) in brain samples from *C9orf72* patients (*Donnelly et al., 2013*). Both RNA binding proteins as a general class and mRNA binding proteins, more specifically, were overrepresented among mRNAs in samples from the diseased brains (*Table 2* and *Supplementary file 1*).

To then assess whether the branching defects could be due to altered transport granule function, we reasoned that changing the levels of transport granule components might suppress or enhance the dendritic defects. We modulated the levels of fly fragile X mental retardation protein (dFMRP), a component of mRNA transport granules and a local translational regulator (*Dictenberg et al.,*

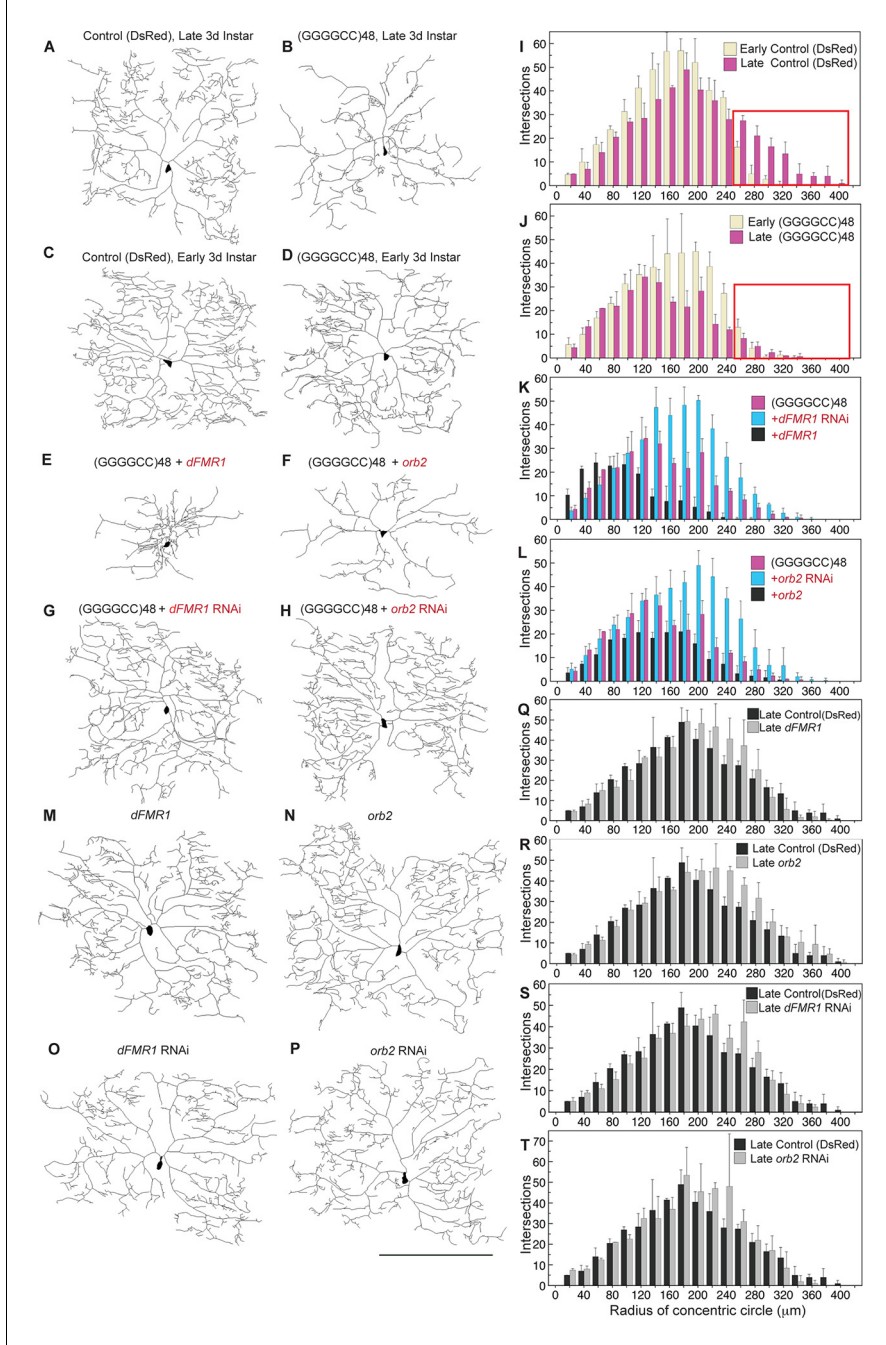

**Figure 4.** (GGGGCC)₄₈-induced dendritic arborization defects are modulated by altered levels of transport granule components in *Drosophila*. (**A–H**) Tracings, from confocal Z-series projections, of the cell body and dendritic arbor of class IV da neurons located in the body wall of *Drosophila* early or late third instar larvae. Expression of (GGGGCC)₄₈ has a dramatic effect on the branching pattern compared with control (a transgene expressing DsRed). The effect on branching is enhanced by upregulation and suppressed by downregulation of *dFMR1* or *orb2*. (**A–D**) GAL4^477 (*Grueber et al., 2003*) driven expression of *UAS-mCD8::GFP* with (**A, C**) *UAS-DsRed* control (*Li et al., 2008*), or with (**B, D**) *UAS-*(GGGGCC)₄₈, in early and late third instar neurons. (**E–H**) GAL4^477 driven expression of *UAS-mCD8::GFP* with (**E**) *UAS-*(GGGGCC)₄₈ and *UAS-dFMR1*, (**F**) *UAS-*(GGGGCC)₄₈ and *UAS-orb2*, (**G**) *UAS-*(GGGGCC)₄₈ and *UAS-dFMR1-RNAi*, or with (**H**) *UAS-*(GGGGCC)₄₈ and *UAS-orb2-RNAi*, in late third instar neurons. (**I–L**) Sholl analysis of traced class IV da neurons shown in (**A–H**), indicating the number of dendrite intersections with circles drawn at increasing radii from the cell body centroid. (**I–J**) Early (yellow) or late (magenta) third instar (**I**) *UAS-DsRed* control or (**J**) *UAS-*(GGGGCC)₄₈ expressing da neurons. Distal intersections (260–400 μm from the cell body centroid) are boxed in red. (**K–L**) Late third instar neurons expressing *UAS-*

*Figure 4 continued*

(GGGGCC)$_{48}$ alone (magenta), or (**K**) with *UAS-dFMR1-RNAi* (cyan) or *UAS-dFMR1* (black), or (**L**) with *UAS-orb2-RNAi* (cyan) or *UAS-orb2* (black). (**M–T**) Expression of the *dFMR1* or *orb2* modifier lines alone minimally alters the dendritic intersection distribution. Controls included comparison of the DsRed control (see A) to *dFMR1* and *orb2* lines in absence of *UAS*-(GGGGCC)$_{48}$. (**M–P**) Tracings of class IV da neurons with *GAL4$^{477}$* driven expression of *UAS-mCD8::GFP* with (**M**) *UAS-dFMR1*, (**N**) *UAS-orb2*, (**O**) *UAS-dFMR1-RNAi*, or (**P**) *UAS-orb2-RNAi*. (**Q–T**) Sholl analysis of traced class IV da neurons shown in (**M–P**). One dorsal neuron from the third or fourth abdominal hemisegment was scored per larvae and three to five larvae were scored per genotype (except for the late third instar control, A; n=2). (**A–H, M–P**) Dorsal, up; anterior, right. The *UAS* constructs did not contain a translation reporter or MS2 tags, see *Figure 1—figure supplement 2B*. Standard deviations are shown. Data are representative of three biological replicates. Bar, 300 μm. See also *Figure 4—figure supplement 1*.

The following figure supplement is available for figure 4:

**Figure supplement 1.** Knock down of *dFMR1* or *orb2* restores (GGGGCC)$_{48}$-induced branching defects.

---

*2008*), and assessed the effects. These studies showed that downregulation of *dFMR1* dramatically mitigated the *UAS*-(GGGGCC)$_{48}$-induced dendritic branching defects (compare *Figure 4G with 4B*) with a near doubling (96% increase) of distal intersections (140–360 μm from cell body; *Figure 4K*). In contrast, upregulation of *dFMR1* in the context of *UAS*-(GGGGCC)$_{48}$ expression potentiated the branching defects, reducing intersections by 70% (120–360 μm from cell body; *Figure 4K*, compare *Figure 4E with 4B*). Studies on a second transport granule component that regulates the local translation of neuritic RNAs, Orb2 (*Cziko et al., 2009*; *Mastushita-Sakai et al., 2010*; *La Via et al., 2013*), showed a similar dramatic modulation of the *UAS*-(GGGGCC)$_{48}$-induced branching defects: distal intersections were doubled upon downregulation (103% increase 140–380 μm from cell body), while upregulation resulted in a 34% overall loss (*Figure 4L*, compare *Figure 4F and H* with 4B). Downregulation of either modifier restored the number of distal intersections (140–400 μm from cell body) compared to the late stage control, to 91% (*dFMR1* RNAi), and to 94% (*orb2* RNAi) (compare *Figure 4G and H to A*; see *Figure 4—figure supplement 1F–G*). Our control experiments indicated that in the absence of *UAS*-(GGGGCC)$_{48}$, up- or downregulation of *dFMR1* and *orb2* resulted in minimal changes in the dendrite intersection distribution. Upregulation of *dFMR1* or *orb2* alone did not reduce the number of intersections (*Figure 4M-N, Q-R*). Knockdown of either *dFMR1* or *orb2* alone resulted in a 3.8 and 6.5% increase in distal dendrite intersections (140–400 μm from the cell body), respectively (*Figure 4O-P, S-T*). The effects of *dFMR1* modulation on da neuron morphology were milder than seen in previous studies, which used null animals, drove expression with a different Gal4 driver, and examined impacts on other specific neurons (*Lee et al., 2003*). Taken together, our data show that expanded GGGGCC microsatellite repeat RNA is present and transported in neurites,

**Table 2.** Expression of transport-granule related transcripts in brains of *C9orf72* patients.

| Expression in *C9orf72* Cortex | Gene subset | Expected percentage | Observed percentage | Chi-squared p-value |
|---|---|---|---|---|
| Upregulated | RNA binding proteins | 7.5 | 15.10 (167/1103) | p<0.0001 |
| | mRNA binding proteins | 3.4 | 7.80 (86/1103) | p<0.0001 |
| | FMRP targets | 4.2 | 4.90 (54/1103) | p=0.2568 |
| | STAT5B targets | 3.1 | 6.07 (67/1103) | p<0.0001 |
| Downregulated | RNA binding proteins | 7.5 | 4.93 (129/2618) | p<0.0001 |
| | mRNA binding proteins | 3.4 | 1.72 (45/2618) | p<0.0001 |
| | FMRP targets | 4.2 | 3.40 (89/2618) | p=0.0389 |
| | STAT5B targets | 3.1 | 1.38 (36/2618) | p<0.0001 |

Comparison of uniquely-identified protein coding genes that were either up- or down-regulated in cortical samples from *C9orf72* patients (*Donnelly et al., 2013*) with transcripts associated with the regulation of RNA. *Supplementary file 1* lists the RNA binding proteins upregulated and downregulated, as noted above.

and that modulation of levels of transport granule components impacts the neuritic defects induced by the expanded microsatellite repeat RNA *in vivo* in *Drosophila*.

## Misregulation of transport granule components in iPSNs from GGGGCC microsatellite expansion carriers

The presence of expanded GGGGCC repeat RNA in transported granules and dramatic modulation of expanded GGGGCC repeat RNA toxicity by *dFMR1* raised the possibility of a functional association between the repeat RNA in the RNA-granules and FMRP protein. In rat spinal cord neurons, both endogenous and exogenous FMRP colocalized in neuritic granules with $(GGGGCC)_{48}$-MS2 and $(CAG)_{100}$-MS2 repeat RNAs in neuronal processes (*Figure 5A–C*, *Figure 5—figure supplement 1A–F*). Thus, association with FMRP was a property of multiple expanded repeat RNAs. Consistent with this observation, both FMRP as well as its interaction partners FXR1 and FXR2, have been shown to interact with GGGGCC RNA repeats by assays that include pull down and proteome arrays (*Almeida et al., 2013*; *Donnelly et al., 2013*; *Haeusler et al., 2014*; *Rossi et al., 2015*).

To illuminate potential functional consequences of the association of the GGGGCC repeat with FMRP in cytoplasmic neuritic RNA granules, we examined whether FMRP-target genes were misregulated in samples derived from human *C9orf72* expansion patients. However, we found no evidence of altered transcript levels of FMRP target genes (*Darnell et al., 2011*) in cortical samples from *C9orf72* patients (*Donnelly et al., 2013*) (*Table 2* and *Supplementary file 1*), which is consistent with a role for FMRP in post-transcriptional gene regulation. We therefore assayed the levels of an FMRP target protein, postsynaptic density protein (PSD-95) (*Todd et al., 2003*; *Muddashetty et al., 2007*; *Zalfa et al., 2007*; *Tsai et al., 2012*), as a readout of FMRP translation regulation in iPSNs derived from two GGGGCC repeat expansion carriers. We found an 89–123% increase in the number, but not the size, of PSD-95 foci per neuron in iPSNs derived from GGGGCC repeat expansion carriers compared to controls (*Figure 5D-E,J*). We also saw a 50–76% increase in total PSD-95 levels (*Figure 5D-E,K*). These results contrast with those obtained when scoring exclusively neuritic PSD-95, for which a change in PSD-95 neuritic puncta was not seen (*Almeida et al., 2013*). We also examined the protein levels of FMRP (which is subject to self-regulation at the mRNA level [*Ashley et al., 1993*]), and found a 51–130% increase in FMRP in patient-derived iPSNs compared with controls (*Figure 5F-G,K*). These results suggest that regulation of FMRP targets could be aberrant in iPSNs derived from *C9orf72* GGGGCC repeat expansion carriers.

We also analyzed a second transport granule component and local translation regulator, human CPEB3 (*Huang et al., 2006*; *Darnell and Richter, 2012*). CPEB3 is a homolog of *Drosophila* Orb2 that modulates GGGGCC repeat toxicity in flies (see *Figure 4*), and is also present in FMRP granules (*Ferrari et al., 2007*) and postsynaptic densities (*Huang et al., 2006*). Total CPEB3 levels were elevated 59–118% in iPSNs with a GGGGCC repeat expansion compared to controls (*Figure 5H-I,K*). The upregulation correlated with a 60–89% increase in CPEB3 foci per neuron in carriers versus controls; foci size was not affected (*Figure 5H-I,J*). The change in CPEB3 could be an independent effect of the toxic RNA or could be a consequence of FMRP-induced changes. Because we found no FMRP enrichment in the nuclei in carrier iPSNs (*Figure 5L*), our data do not support a nuclear GGGGCC repeat RNA-mediated FMRP sequestration model. To investigate the functional significance of dysregulated CPEB3 levels, we asked whether its target genes might be misexpressed. CPEB3 can modulate expression of targets of the transcription factor STAT5B (*Peng et al., 2010*). Consistent with disruptions in CPEB3/STAT5B-modulated transcription, cortical samples from *C9orf72* patients (*Donnelly et al., 2013*) exhibit misregulation of STAT5B target genes (*Kanai et al., 2014*) (*Table 2*). These results indicate that expanded GGGGCC repeat RNA may interfere with the local translation machinery and indirectly modify transcriptional programs. Together, these data suggest that expanded microsatellite repeat RNAs like GGGGCC that are incorporated into granules within neurites may have local effects that contribute to neurodegeneration.

## Discussion

Here we have identified a novel function of expanded microsatellite RNA repeats in conferring neuritic RNA granule localization. Our data indicate that expanded repeat RNAs with specific structural context (e.g. stem-loop for CAG, CUG, and CCUG, and G-quadruplex and stem-loop for GGGGCC repeat RNA) can be recognized by the mRNA localization machinery, can become incorporated into

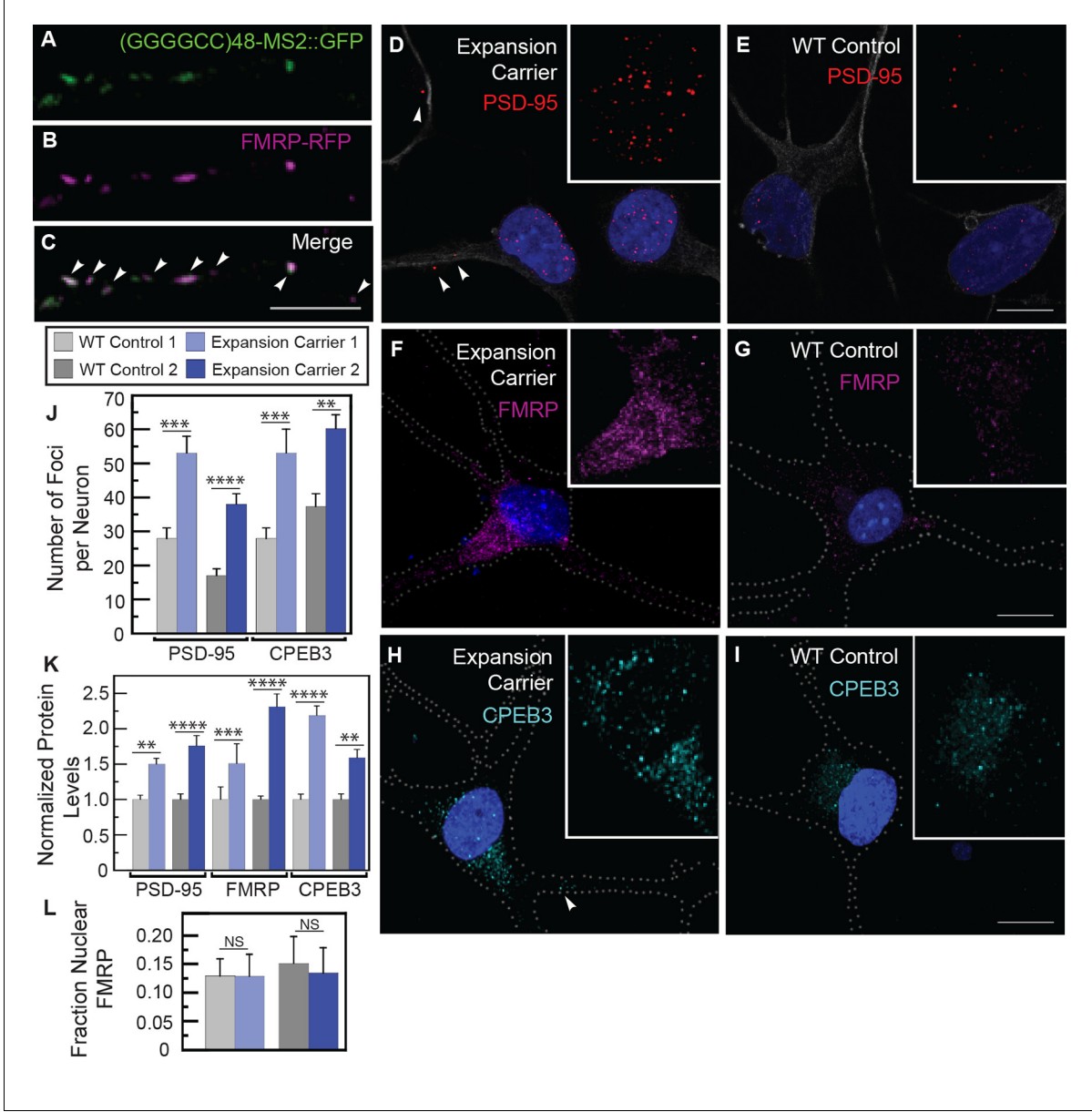

**Figure 5.** Misregulation of transport granule components in human iPSNs from carriers with a *C9orf72* GGGGCC expansion. (A–C) Neuritic particles consisting of expanded GGGGCC repeat RNA co-label for FMRP. Rat primary spinal cord neurons were transfected with NLS-CP-GFP (green), FMRP-RFP (magenta), and (GGGGCC)$_{48}$-MS2, and neuronal processes were defined as regions of interest. Colocalization coefficients M1 (FMRP-RFP overlap with (GGGGCC)$_{48}$-MS2 RNA) and M2 ((GGGGCC)$_{48}$-MS2 RNA overlap with FMRP-RFP) were 0.64 ± 0.15 SD and 0.68 ± 0.23 SD, respectively (n=6 neurons). Colocalization coefficients for overlap between endogenous FMRP and (GGGGCC)$_{48}$-MS2 were M1=0.61 ± 0.06 SD and M2=0.56 ± 0.14 SD (n=5 neurons; not shown). See *Figure 1—figure supplement 2A* and Materials and methods for construct details. Data are representative of three biological replicates. Confocal Z-series projections. (D–K) FMRP targets (D–E) PSD-95 and (F–G) FMRP, as well as (H–I) CPEB3, a local translation regulator, are increased in human iPSNs from *C9orf72* GGGGCC expansion carriers, with a concomitant increase in PSD-95 and CPEB3 foci. High magnification of cell bodies are shown as insets. Neurites were marked with α-b III Tubulin or are outlined (dotted line), and (D) neuritic PSD-95 foci are indicated (arrowheads). (D–I) Images for GGGGCC expansion carrier 2 are shown. (J–L) Key for carriers and controls is shown at top. (J–K) Quantitation of PSD-95 and CPEB3 (J) foci, and of (K) total protein levels by immunostain in human iPSNs from carriers with a *C9orf72* GGGGCC expansion. Kruskal–Wallis analysis for carrier versus control for all conditions: *p* < 0.0001. Post-hoc Dunn's test, multiplicity adjusted p-values: ****p < 0.0005; ***p < 0.0015; **p < 0.018; N.S. *p* > 0.05. (J) From left to right, PSD-95: control, n=812 foci in 29 neurons; carrier 1, n=1590 foci in 30 neurons; control, n=851 foci in 49 neurons; carrier 2, n=1455 foci in 38 neurons. CPEB3: control, n=980 foci in 35 neurons; carrier 1, n=2067 foci in 39 neurons; control, n=735 foci in 21 neurons; carrier 2, n=1980 foci in 44 neurons. (K) From left to right, PSD-95: control, n=29; carrier 1, n=30; control, n=37; carrier 2, n=48 neurons scored. FMRP: control, n=32; carrier 1, n=30; control, n=25; carrier 2, n=27 neurons scored. CPEB3: control, n=34; carrier 1, n=37; control, n=17; carrier 2, n=49 neurons scored. (L) FMRP is not sequestered in the nuclei of carrier iPSNs. Quantitation of the fraction nuclear to total FMRP in carrier vs. control iPSNs.

*Figure 5 continued on next page*

*Figure 5 continued*

Data are averages from carrier 1, carrier 2, and controls ± standard error of the mean. From left to right: control, n=5; carrier 1, n=5; control, n=8; carrier 2, n=6 neurons scored. All comparisons are non-significant by ANOVA and post-hoc Sidak's t-test. Confocal Z-series projections are shown. Bars: (E, G, I) 10 μm; (C) 5 μm. DAPI: blue. See also *Figure 5—figure supplement 1*. ANOVA, analysis of variance; CP-GFP, MS2 RNA-binding coat protein fused with green fluorescent protein; FMRP, fragile X mental retardation protein; NLS, nuclear localization signal; PSD, postsynaptic density protein; RFP, red fluorescent protein.

The following figure supplement is available for figure 5:

**Figure supplement 1.** FMRP colocalizes with neuritic (GGGGCC)$_{48}$-MS2 and (CAG)$_{100}$-MS2 RNA.

neuritic RNA transport granules, and, at least for expanded GGGGCC hexanucleotide repeat RNA, may disrupt RNA granule function. The RNAs expressed are directed to the cytoplasm with a poly(A) tail, as are repeats that occur within the mRNA of the respective disease genes. In the case of the hexanucleotide expansion in C9orf72, although the repeat is defined as intronic, we saw neuritic GGGGCC RNA granule localization in iPSNs, indicating the repeat can localize to the cytoplasm in disease. Notably among the large portion of mRNAs that are localized, RNAs with stem-loop structure commonly function as cis-acting localization signals (*Ferrandon et al., 1994*; *Serano and Cohen, 1995*; *Cohen et al., 2005*; *Snee et al., 2005*; *Van De Bor et al., 2005*; *Dienstbier et al., 2009*). Indeed, in flies, 7 transcripts have been demonstrated to localize through minus-end directed transport along microtubules, and these mRNAs all contain one or more stem-loops within their localization signal. Although not similar to each other at the primary sequence level, all of these localization signals are recognized by the same localization machinery (*Dienstbier et al., 2009*). In addition, G-quadruplex consensus RNA sequences have also been shown to be cis-acting elements that are both necessary and sufficient for neuritic localization of PSD-95 and CaMKIIα, two dendritically localized mRNAs (*Subramanian et al., 2011*). Indeed, about one-third of the best characterized dendritic mRNAs contain a putative G-quadruplex in their 3'UTRs (*Subramanian et al., 2011*; *Stefanovic et al., 2015*). Hence, not only do G-quadruplex consensus sequences and disease-associated GGGGCC repeat RNA assume G-quadruplex structure, these RNAs also appear to have a similar common function as neuritic localization signals. These observations underscore the findings we report that, structured RNAs, like CAG and GGGGCC, are localizing to dynamic neuritic granules.

We find that neuritic localization of the expanded GGGGCC hexanucleotide repeat RNA occurs in association with neuritic defects. Neurons with expanded GGGGCC RNA granules in neurites have a decrease in primary branches compared with controls. We do not see a comparable decrease when the expanded RNA is localized merely to the soma, or when it is present in nuclear foci, consistent with a recent report (*Tran et al., 2015*). Importantly, the branching defects associated with the neuritically localized expanded repeat are not seen with a similarly localized non-expanded (GGGGCC)$_3$ repeat—branching in neurons with neuritic non-expanded repeat is not significantly different from branching in neurons with somatic or nuclear expanded repeat. These data indicate that the incorporation of expanded microsatellite repeat RNAs into granules within neurites induces dysfunction. The finding of branching defects in rat primary spinal cord neurons in culture was also extended to da neurons in *Drosophila*. *In vivo*, the dendritic arbors of da neurons are normal early, but later show a different pattern with fewer intersections and smaller field. This effect is distinct from defects in endosomal transport, as described for dynein loss-of-function mutations (*Satoh et al., 2008*), indicating it is unlikely to be due to vesicular traffic transport defects. Furthermore, two transport granule components (FMRP and Orb2, the fly CPEB3 ortholog) are novel modifiers of GGGGCC toxicity. Our studies also provide evidence that the expanded GGGGCC repeat RNA may compromise local translation regulation: the FMRP targets, PSD-95 and FMRP, appeared present at elevated levels in iPSNs from *C9orf72* hexanucleotide expansion carriers. GGGGCC repeat RNA could disrupt FMRP-mediated translational repression or increase FMRP-mRNA target stability, the latter scenario being less likely because PSD-95 mRNA levels are similar in carrier vs. control iPSNs (*Almeida et al., 2013*). In FMRP knockout mice, PSD-95 mRNA is destabilized and PSD-95 levels reduced (*Zalfa et al., 2007*; *Zhu et al., 2011*)—similarly, knockdown of FMRP might lead to destabilization of its mRNA targets, thus counteracting translational derepression by toxic GGGGCC repeat RNA in disease. There are a large number of mRNAs regulated by FMRP and

CPEB3, many or all of which may factor into neurotoxicity. Consistent with our observation that CPEB3 protein levels are upregulated in iPSNs from GGGGCC expansion carriers, we find CPEB3/STAT5B-regulated genes are dysregulated in samples from *C9orf72* patient cortex (*Donnelly et al., 2013*), see *Table 2*).

The mechanisms of toxicity or pathogenesis of expanded microsatellite repeat RNA include protein translation and sequestration of binding proteins. A study in the fly showed that RAN translation products generated from the GGGGCC repeat RNA can be toxic. Moreover, they found that a GGGGCC repeat is toxic *in vivo*, but toxicity is minimal if the sequence is not a pure GGGGCC, but is interrupted by stop codons (and thus could not code for peptides) (*Mizielinska et al., 2014*). However, alterations in the RNA sequence required to block RAN translation (introduction of stop codons) may well interfere with the intricacy of RNA-protein interactions, such as those required for subcellular RNA localization, and/or those that mediate toxicity. Mechanisms beyond RAN translation may well contribute to neurodegeneration conferred by expanded GGGGCC repeat RNA. Targeting of expanded microsatellite repeat RNA to the neuritic granules that we document may disrupt local mRNA translation, and might also interfere with proper trafficking of cellular RNAs. We speculate that the presence of microsatellite repeat RNA in neurites might also result in local RAN translation, and that RAN translation products in the neuritic subcellular compartment could contribute to neurite loss.

Our data support a novel model in which neuritically localized expanded microsatellite repeat RNAs associate with neuritic RNP granule components and disrupt their function, resulting in neuritic defects. This mechanism may contribute to ALS/FTD disease in patients bearing the GGGGCC repeat expansion, as we have shown strong effects in iPSC-derived neurons from GGGGCC expansion carriers, in cultured rat spinal cord neurons, and *in vivo* in a *Drosophila* model. In culture, we have shown many different expanded microsatellite repeat RNAs are incorporated into neuritic granules, and at least several are actively transported. For the GGGGCC repeat, a number of proteins that bind the repeat (hnRNP A3) or are modifiers of GGGGCC repeat toxicity (Pur alpha and hnRNP A2/B1) are implicated in transport granule function (*Jin et al., 2007*; *Sofola et al., 2007*; *Xu et al., 2013*). Interestingly, mutations in TDP-43 impair neuritic mRNA transport in primary and stem-cell derived neurons and are causative of ALS (*Alami et al., 2014*); TDP-43 pathology also characterizes many repeat expansion diseases (*Elden et al., 2010*; *Toyoshima and Takahashi, 2014*). Thus, multiple lesions could converge at the functional level to result in disrupted mRNA transport granule function.

## Materials and methods

### Plasmids and gene synthesis

RFP-DCP1, DsRed, and pGW were from Dr. Robert Kalb (Department of Pediatrics, University of Pennsylvania School of Medicine), and FMRP-RFP was a kind gift from Dr. Ian Macara (Department of Cell and Developmental Biology, Vanderbilt University). A backbone was designed to receive the repeat sequences $(CAG)_{40}$, $(CAG)_{70}$, $(CAG)_{100}$, $(CUG)_{100}$, $(CCUG)_{100}$, $(GGGGCC)_{48}$, and $(GAA)_{100}$. The backbone, as well as the repeat sequences, were synthesized and ligated into pUC57 (GenScript, Piscataway, NJ). The repeat sequences contained 5′ EcoRI and 3′ BamHI sites, and the first base of the first tandem repeat was omitted if it started with cytosine. The backbone contained the following in 5′ to 3′ order: a 6Stop sequence (carrying six 5′ stop codons (underlined) in the leader sequence, two in each reading frame) containing a 3′ EcoRI site (<u>TAG</u>C<u>TAG</u>G<u>TAA</u>CT<u>AA</u>G<u>TAA</u>CTAGAATTC (*Renton et al., 2011*)), followed by a BamHI site (GGATCC), then by sequences encoding FLAG-, HA-, and Myc-tags (AGGATTACAAGGACGACGACGACAAGTAGCTACCCATACGACGTTC-CAGATTAC CTTAACGAACAGAAACTCATCTCTGAAGAGGATCTGAACATGCATACGGGTCATC T-CACCATCACCACTAATAGATAGTGAATAATGAATTTAAATTAATAGATAGTGAATA TGA), and then 12 MS2 stem-loops (*Haim-Vilmovsky and Gerst, 2009*) (of sequence (CCTAGAAAACATGAGGATC-ACCCATGTCTGCAGGTCGACTCTAGAAAACATGAGGATCACCCATGTCTGCAG TATTCCCGGGT-TCATTAGATCCTAAGGTACCTAATTG)5 CCTAGAAAACATGAGGATCACCCATGTCTGCAG GTCGACTCCAGAAAACATGAGGATCACCCATGTCTGCAG TATTCCCGGGTTCATT CTCGAG AGATCT). The backbone was then cloned into pGW using external restriction sites and the repeat sequences were then inserted between EcoRI and BamHI restriction sites of the backbone. $(CAG)_{20}$-

MS2 and (GGGGCC)$_3$-MS2 were made by polymerase chain reaction (PCR) using complimentary oligos and ligated into pGW containing the backbone, as described above. LacZ was amplified by PCR and ligated into pGW-MS2 to generate LacZ-MS2. (CAG)$_{100}$, shown in *Figure 1D–E* and in *Figure 1—figure supplement 1A–B*, was inserted into 6Stop-FLAG-HA-Myc to generate 6Stop-(CAG)$_{100}$-FLAG-HA-Myc and cloned into pcDNA, and lacked an MS2 tag. CP-(GFP)$_2$ (*Haim-Vilmovsky and Gerst, 2009*) with a 5' NLS or nuclear export signal (NES) sequence was cloned into pGW. See *Figure 1—figure supplement 2A* for construct diagrams.

## Neuron culture and immunostain

Embryonic Sprague Dawley rat spinal cord neurons from embryonic day 14 were grown on previously established cortical postnatal d1-3 astrocyte monolayers (*Mojsilovic-Petrovic et al., 2006*). Neurons were grown for 5 d before being transfected with Lipofectamine 2000 (Invitrogen, Carlsbad, CA), according to the manufacturer, and using a 1:3:3 ratio of NES- or NLS-CP-GFP, MS2 tagged sequences, and DsRed or RFP plasmids, respectively. Neurons were fixed at 17–24 hr post transfection and processed according to standard procedures. Antibodies were added overnight at 4°C and included chicken α-GFP (1:2000; A10262, Invitrogen, Carlsbad, CA), mouse α-mRFP (1:2000; ab65856, Abcam, Cambridge, MA). Mouse α-FMRP (clone 2F5-1; *Christie et al., 2009*) was added after steam antigen retrieval. Neurons on coverslips were mounted in Vectashield Mounting Medium with 4',6-diamidino-2-phenylindole (DAPI; Vector Laboratories, Burlingame, CA). A minimum of three independent transfections with experimental samples along with controls were performed for all samples and yielded similar results across biological replicates. Fibroblast-derived iPSNs from GGGGCC hexanucleotide expansion carrier 1 (line #5) and carrier 2 (line #11), and from control lines (#17 and #20) (*Almeida et al., 2013*) were fixed, and stained using mouse α-FMRP as above, mouse α-PSD-95 (1:200; 6GG-IC9, Pierce/Fisher, Rockford, IL), rabbit α-CPEB3 (1:200; ab10883, Abcam, Cambridge, MA), or chicken α-b III Tubulin (1:1000; AB9354, Millipore, Billerica, MA). Due to the extensive nature of required quantitation, each carrier was independently experimentally analyzed with a control. All iPSC lines were grown and differentiated to neurons in parallel.

## *In situ* hybridization

DIG labeled (CUG)$_8$ and (CAG)$_8$ sense and antisense oligonucleotide probes were generated (IDT DNA, Coralville, IA), *in situ* hybridization was performed (*Wilk, 2010*), and the probe signal was amplified with the Tyramide Signal Amplification system (Perkin Elmer, Whaltham, MA) using a fluorescein kit according to the manufacturer. A Cy3-conjugated (GGCCCC)$_4$ oligonucleotide probe was used for *in situ* hybridization of iPSNs as described (*Almeida et al., 2013*). A Leica confocal microscope equipped with a HyD detector was used for detection of GGGGCC RNA particles.

## *Drosophila* da neuron analysis

Early and late third instar larvae were filleted in ice cold phosphate-buffered saline (PBS), fixed in PBS/4% paraformaldehyde, and stained with chicken α-GFP as described above. Post-fix and post-stain washes included three rinses and 3 × 15 min in PBS/0.3% Triton X-100. Secondary antibodies were conjugated to Alexa 488 (Invitrogen, Carlsbad, CA).

## *Drosophila* strains

The *Drosophila* lines to knock down *orb2* (genotype y$^1$ v$^1$; P{TRiP.JF023076}attP2), and *dFMR1* (genotype y$^1$ sc* v$^1$; P{TRiP.HMS00248}attP2) were from the Bloomington stock center. The *UAS*-(GGGGCC)$_{48}$ repeat sequence (with the 6Stop sequence but without the translation or MS2 tags noted above; see *Figure 1—figure supplement 2B*) was subcloned into pUAST to generate *UAS*-(GGGGCC)$_{48}$ and the construct was injected to generate transgenic strains (Genetic Services, Inc., Cambridge, MA). *UAS-dFMR1* was from Dr. Thomas Jongens (Department of Genetics, University of Pennsylvania School of Medicine). The *UAS*-DsRed strain was used as a control for *UAS*-(GGGGCC)$_{48}$. *UAS*-DsRed (*Bilen and Bonini, 2007*) and *UAS-orb2* are described (*Dictenberg et al., 2008*).

## Microscopy

Images of rat and da neurons were captured on a Leica TCS SP5 confocal microscope and processed with the Leica Application Suite (LAS) software (Leica Microsystems, Wetzlar, Germany). Sequential acquisition was applied when capturing an image in multiple channels. Similar voltage settings were applied when capturing images of rat neurons transfected with different constructs, and a saturation threshold was applied. For *Figure 1*, above-background fluorescence that was clearly discernable by eye as having a clear particle limit was scored as a particle.

## Live imaging

Images of spinal cord neurons at 17–24 hr post transfection, were collected with a Deltavision Core Deconvolution Microscope (Applied Precision, Issaquah, WA), equipped with an Olympus IX70 microscope and a Photometrics CoolSNAP HQ camera, a 60X, 1.42 NA oil immersion PlanApo lens (Olympus, Tokyo, Japan), and softWoRx (Applied Precision, Issaquah, WA) acquisition software. Environmental control was provided by a home-built plexiglass cage surrounding the entire microscope, kept at 37°C and 5% $CO_2$. Individual frames were generated at 1 s intervals for single channel imaging. The percentage of neurons with distal $(GGGGCC)_{48}$-MS2, and $(CAG)_{100}$-MS2 RNA particles detected by live imaging (six sessions for $(GGGGCC)_{48}$-MS2, and two sessions for $(CAG)_{100}$-MS2 was similar to that seen in fixed neurons from multiple biological replicates. Two sessions for $(GGGGCC)_{48}$-MS2 were excluded due to low transfection efficiency.

## Data analysis

Videos were generated and particles were tracked manually with Fiji software. Quantitative colocalization analysis was performed using Volocity software version 6.2.1 (Perkin Elmer, Whaltham, MA). The colocalization coefficients (M1 and M2) were computed (*Manders et al., 1993*) for regions of interest (ROI). These ROIs were manually selected to only target neuronal processes. We analyzed >5 neurons for each condition to ensure that M1 and M2 were similar when comparing cells within the same sample and between distinct biological replicates (>3). The intensity thresholds for the colocalization coefficients were determined using an auto-threshold method (*Costes et al., 2004*). Spinal cord neurons and *Drosophila* da neurons were traced, and the tracings were analyzed with Neurolucida and Neuroexplorer software, respectively (MicroBrightField, Colchester, VT). For the rat dendritic arbor analysis, only neurons that had a cell body diameter of >20 μm, and had more than two primary arbors were included. A pre-established standard cell sample size (n≥20; Drs. Lei Zhang and Robert Kalb, personal communication) was used for this type of analysis, except for samples that had nuclear or neuritic $(GGGGCC)_{48}$-MS2 RNA (n=12), due to the limiting inclusion criteria used. For quantitation of PSD-95 and CPEB3 particles in iPSNs, z-stacks taken with a 63× objective acquired on a Leica confocal microscope equipped with a HyD detector were projected, and the cell body was outlined. The particle number and size were analyzed using the 'analyze particle' function of Image J (NIH), using the Yen or Max Entropy auto-thresholding methods. Our analysis of PSD-95 in the entire cell body differs from previous quantitation solely in dendrites (*Almeida et al., 2013*). Total protein levels of FMRP, PSD-95, and CPEB3 in iPSNs was measured by outlining the entire neuron using Image J. For analysis of nuclear FMRP the nucleus (based on DAPI stain), and the whole neuron were outlined, measured using Image J, and the nuclear intensity was divided by the total neuron intensity. For *Figure 3I and J*, expression levels were measured using Image J to calculate the mean intensity; the cell bodies of the neurons were selected as the ROI for these analyses. To determine the somatic expression levels, signal from the nucleus, defined by DAPI staining, was subtracted.

## Sample randomization

All samples, including all animal experiments, were randomly assigned to processing order, and for cell transfections, the positions in the wells were random. Data was also collected randomly.

## Statistical analysis

Statistical tests were performed using R 3.1.2 (*Figure 1*) or Prism 6 software from Graphpad, La Jolla, CA (*Figures 3* and *5*, and *Table 2*). For *Figure 1L–M*, the data were expressed as a binomial, with cells categorized as having neuritic RNA or not. Within each condition (for example, construct

or carrier/control), the cells were grouped by experiment to account for potential variability between experiments. The data were fitted with a log-linear generalized linear model in R 3.1.2 (Pumpkin Helmet) using the glmer function of the lme4 package, with post-hoc analyses comparing each construct/condition to control. In the case of *Figure 1M*, where the LacZ-MS2 construct had no variance, the model could not converge. Therefore a single LacZ-MS2 data point was switched from nuclear to neuritic. The same operation was done for control iPSNs in *Figure 1L*. Both of these changes were conservative as they were in the opposite direction of the observed effect. For the analyses in *Figure 5*, the Brown–Forsythe test indicated that the samples exhibited different variances. We therefore conducted nonparametric tests for these analyses. For the analysis of gene lists, uniquely identifiable, well-annotated protein coding transcripts (i.e., those with a refseq identifier beginning with 'NM') that were misregulated in *C9orf72* patient samples were compared with several gene lists as indicated in the text: RNA binding proteins and mRBP classifications were from (*Gerstberger et al., 2014*); FMRP targets were GSE45148 from (*Darnell et al., 2011*); *C9orf72* targets were from (*Donnelly et al., 2013*); Stat5b targets were from (*Kanai et al., 2014*). The percent of *C9orf72*-regulated transcripts was compared with the percentage expected by chance given the prevalence of the RNAs in the ~20,000 protein-coding transcripts present in the human genome using a chi-squared analysis with a significance threshold of $p=0.01$.

## Acknowledgements

We are grateful to Drs. William Motley, Thomas Jongens, and Ian Macara for generously sharing reagents. We thank Dr. Amin Ghabrial for discussion, and Drs. Yongqing Zhu, Joshua Black, Mike O'Connor, Jelena Mojsilovic-Petrovic, and Lei Zhang for technical assistance. We thank the patients and their families who generously supported this work. This work received funding from an NIH NINDS NRSA (F32-NS067902 to ASB), an NIH training grant (ASB, T32-AG000255, to Virginia M-Y Lee), and the NIH (R01-NS079725 to F-BG, R21-AG042179, R21-NS077909 and R01-NS052325 to RK, R00-MH090237 to MRA, and R01-NS0736690 to NMB).

## Additional information

### Funding

| Funder | Grant reference number | Author |
| --- | --- | --- |
| National Institute of Neurological Disorders and Stroke | F32NS067902 | Alondra Beatriz Schweizer Burguete |
| National Institute on Aging | T32AG000255 | Alondra Beatriz Schweizer Burguete |
| National Institute of Neurological Disorders and Stroke | R01NS079725 | Fen-Biao Gao |
| National Institute on Aging | R21AG042179 | Robert Kalb |
| National Institute of Neurological Disorders and Stroke | R21NS077909 | Robert Kalb |
| National Institute of Neurological Disorders and Stroke | R01NS052325 | Robert Kalb |
| National Institute of Mental Health | R00MH090237 | Michael R Akins |
| National Institute of Neurological Disorders and Stroke | R01NS073660 | Nancy M Bonini |

The funders had no role in study design, data collection and interpretation, or the decision to submit the work for publication.

## Author contributions
ASB, Conceived and designed the experiments, acquired the data, analyzed and interpreted the data, wrote and revised the manuscript; SA, Input into select approaches in the conception and design of experiments, input into select approaches of acquiring the data, input into the manuscript; F-BG, RK, Input into select approaches in the conception and design of experiments, input into the manuscript; MRA, Input into select approaches in the conception and design of experiments, input into select approaches of acquiring the data, input into interpretation of the data, input into the manuscript; NMB, Conceived and designed the experiments, input into interpretation of the data, wrote and revised the manuscript

## Ethics
Animal experimentation: The studies with animal tissue were performed in strict accordance with the recommendations in the Guide for the Care and Use of Laboratory Animals of the National Institutes of Health. All of the animals were handled according to approved institutional animal care and use committee (IACUC) protocols (#597) of Children's Hospital of Philadelphia. The human stem cell studies were performed with approval by the Institutional Biosafety Committee, protocol number I-435-10, of the University of Massachusetts Medical School, Worcester, MA.

## Additional files

### Supplementary files
• Supplementary file 1. Genes related to RNA granules that are misregulated in brain tissue from *C9ORF72* patients. Lists of the RNA binding proteins upregulated or downregulated in cortical samples from *C9orf72* patients (*Donnelly et al., 2013*). Analysis is presented in *Table 2*.

### Major datasets
The following previously published datasets were used:

| Author(s) | Year | Dataset title | Dataset URL | Database, license, and accessibility information |
|-----------|------|---------------|-------------|--------------------------------------------------|
| Darnell J, van Driesche SJ, Zhang C, et al. | 2011 | HITS-CLIP analysis of FMRP mRNA binding sites from P11-P25 mouse brain polysomes | http://www.ncbi.nlm.nih.gov/geo/query/acc.cgi?acc=GSE45148 | Publicly available at the NCBI Gene Expression Omnibus, Accession No. GSE45148 |

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
