## [Decision Letter]

Thank you for submitting your work entitled "GGGGCC microsatellite RNA is neuritically localized, induces branching defects, and perturbs transport granule function" for peer review at *eLife*. Your submission has been evaluated by a Senior editor, a Reviewing editor (Robert H Singer), and two reviewers (Roy Parker and Harry Orr).

The reviewers have discussed the reviews with one another and the Reviewing editor has drafted this decision to help you prepare a revised submission.

Summary:

This manuscript addresses the possible mechanisms by which microsatellite expansions in mRNAs may lead to neurodegenerative disease. The main contribution is to show that GGGGCC and CAG repeat RNAs can localize to granules within neurites, which is clearly documented. Additional work leads the authors to argue that repeat RNAs in neurites leads to loss of dendritic branching and toxicity, which may be related to changes in the levels of key components of RNP granules.

Essential revisions:

Roy Parker raises some important points that would need to be addressed, primarily stronger data showing that the repeats are causative for both localization and toxicity (see complete comments below). If you think that you can substantially address these comments within the next few months, we would be willing to consider a revised manuscript.

Additional points:

Additional comments from Harry Orr: “I think reviewer 1 (Roy Parker) raised some important points (1-4). Demonstrating neurite pathology in disease, I assume in humans, is a tall order particularly since there is not a mouse model of at least C9ORF except for the viral mediated disease recently published in Science”.

The Reviewing editor also consulted another expert, who provided the following comments: “The only novelty in this paper is the existence of GGGGCC RNA foci in dendrites (whereas other *C9orf72* papers show RNA foci only in cell body or nucleus). The protein partner of GGGGCC RNA foci they found was FMRP, which unfortunately for these authors, is now published by another group (Rossi et al. J Cell Sci 2015). Although the binding between GGGGCC and FMRP is not novel anymore, they showed nice rescue with FMRP knockdown in flies, suggesting functional importance of this interaction”.

*Reviewer #1:*

This manuscript addresses the possible mechanisms by which microsatellite expansions in mRNAs may lead to neurodegenerative disease. The main contribution is to show that GGGGCC and CAG repeat RNAs can localize to granules within neurites, which is clearly documented. Additional work leads the authors to argue that repeat RNAs in neurites leads to loss of dendritic branching and toxicity, which may be related to changes in the levels of key components of RNP granules.

The strength of the manuscript is to show repeat RNAs in neurites, and to suggest possible phenotypic consequences of that localization. The limitations of the work are: 1) clear documentation that the neurite localization is specific to the repeat sequences, 2) that the toxicity is due to neurite localization of RNAs, and 3) some limitations in specific experiments. I would think the manuscript should be improved technically before further consideration. A downstream issue will be whether this type of toxicity, without demonstration in disease, or an understanding of mechanism will be sufficient for publication in *eLife*.

Specific Comments:

1) It is clear that the repeat RNAs can be seen in neurites. However, the basis for this specificity is not clear. Is the control the exact same RNA without the repeats inserted? From my reading of Methods it seems not. Would any highly expressed RNA end up in neurites? Evidence that the repeats are what allows accumulation should be either provided, or if already present, clarified by more explicit description of the experiments and control RNAs examined. A simple figure showing the compared RNAs would be helpful.

I think it would also improve the clarity and strength of this experiment to describe the cells more completely, e.g. what percentage of cells has nuclear foci? Somatic foci? Neurite foci? Does neurite foci correlate with expression level in the soma? If not, showing so would improve the rigor of the experiment.

2) A second key contribution is the conclusion that neurite RNA limits arborization. To make this rigorous it would be important to again show that this only correlates with neurite RNA and not soma foci, nuclear foci, or expression level. Since one is showing correlations, being sure to examine if there are other possible correlations is important.

3) It was unclear how the authors are concluding there are changes in the levels of mRNP granule components. As I read the work, it seems they are showing a change in average intensity in granules when the repeat RNA is expressed. Is this due to less protein in the cells? To less granule formation? This should be clarified.

4) The presentation of the effects on neurite arborization is difficult to follow. I request the following alterations:

A) For the experiments in rat neurons it would be appropriate to examine each neuron examined for the overall level of the repeat RNA, the presence of soma foci and nuclear foci. This is an attempt to show that the correlation with neurite foci is at least as strong a correlation as one can assess.

B) For the experiments in flies I request the following:

i) A clear description of the differences between the "control" RNA and the (GGGGCC)_48_ RNA. Are they the same mRNAs just with an insertion of repeats? Or are they actually different? (dsRed vs. just a repeat RNA?) From my reading of Methods it seems they are different RNAs.

ii) For comparison of the effects of the (GGGGCC)_48_ expansion, the key comparison is the control to the (GGGGCC)_48_ expression lines at early and late time. The figures would be easier to interpret if Figure 4 was comparison of early time points for control and (GGGGCC)_48_ and Figure 4 was comparison of late control to late (GGGGCC)_48_ RNA.

iii) Earlier work (Lee et al., 2003) had shown that loss of function mutations in FMR result in higher-order dendritic branches, while the overexpression of FMR dramatically decreases dendritic branching. Given this it is surprising that no effects of FMR OE or RNAi are seen in the wild-type control. It might be worth commenting on this and clarifying that they see a different result. I raise this issue since OE of many FMR interacting proteins decreased dendritic branching in similar experiments (Cziko et al., 2009; Barbee et al., 2006). Perhaps a different driver is being used that is not as strong, which should just be clarified?

I would suggest that these key wild-type controls be included in the main text and presented in the same manner with the same type of images as the (GGGGCC)_48_ lines since this is critical to interpretation of the experiment.

*Reviewer #2:*

Since discovery of the C9ORF hexanucleotide repeat as a major cause of inherited ALS/FTD much effort has gone into understanding its pathogenic basis. This study presents data in support of the concept that the repeat, based on its structural features, directs repeat-containing RNA to transport granules in neurons. Interestingly, they show that the ability of the RNA to form a structure is critical for its targeting to RNA granules – for example the Frataxin repeat that does not for a structure did not localize to granules. They go onto demonstrate that association of repeat-RNA to granules is linked to neurite degeneration and alterations in proteins, whose translation is regulated by FMRP and Orb2. Overall this a strong study that uses results from several approaches and model systems to support their novel conclusion that repeat containing RNA disrupts RNA transport granule function that may be another pathogenic mechanism. A minor point that I think should be addressed is:

In the subsection “Neuritic subcellular localization is a common property of highly structured microsatellite repeat RNAs”, the authors describe that the number of granules with wt G4C2 RNA is much higher than those form in the presence of a G4C repeat with 48 repeats. Does this suggest that while targeting information is retained by expanded repeat the ability to form granules is compromised?

---

## [Author Response]

*Essential revisions: Roy Parker raises some important points that would need to be addressed, primarily stronger data showing that the repeats are causative for both localization and toxicity (see complete comments below). If you think that you can substantially address these comments within the next few months, we would be willing to consider a revised manuscript. Additional points: Additional comments from Harry Orr: “I think reviewer 1 (Roy Parker) raised some important points (1-4). Demonstrating neurite pathology in disease, I assume in humans, is a tall order particularly since there is not a mouse model of at least C9ORF except for the viral mediated disease recently published in Science”. The Reviewing editor also consulted another expert, who provided the following comments: “The only novelty in this paper is the existence of GGGGCC RNA foci in dendrites (whereas other C9orf72 papers show RNA foci only in cell body or nucleus). The protein partner of GGGGCC RNA foci they found was FMRP, which unfortunately for these authors, is now published by another group (Rossi et al. J Cell Sci 2015). Although the binding between GGGGCC and FMRP is not novel anymore, they showed nice rescue with FMRP knockdown in flies, suggesting functional importance of this interaction”.*

We thank the external consultant for appreciating the rescue of GGGGCC-induced defects by dFMR1 knockdown, which we agree suggests that the RNA-dFMR1 interaction is functionally important. We likewise agree that a dFMR1 interaction with GGGGCC RNA has been previously described, by in vitropull down and mass spectrometry (Almeida et al., 2013; Rossi et al., 2015).

However neither of these studies shows colocalization between GGGGCC repeat RNA and FMRP. In the Rossi et al. paper, overexpression of (GGGGCC)_31_ RNA induced formation of Purα and FMRP foci in NSC34 and in HeLa cells, but the (GGGGCC)_31_ RNA and FMRP were not found to colocalize, see Figure 4 and Table 2. In our study we show in vivoassociation of GGGGCC repeat RNA and FMRP in rat primary spinal cord neurons by colocalization, and genetic data that supports a functional interaction. Therefore our data significantly extends previous observations, by providing data in support of a physical and functional interaction between the RNA and FMRP.

Reviewer #1:

*The strength of the manuscript is to show repeat RNAs in neurites, and to suggest possible phenotypic consequences of that localization. The limitations of the work are: 1) clear documentation that the neurite localization is specific to the repeat sequences, 2) that the toxicity is due to neurite localization of RNAs, and 3) some limitations in specific experiments. I would think the manuscript should be improved technically before further consideration. A downstream issue will be whether this type of toxicity, without demonstration in disease, or an understanding of mechanism will be sufficient for publication in eLife. Specific Comments: 1) It is clear that the repeat RNAs can be seen in neurites. However, the basis for this specificity is not clear. Is the control the exact same RNA without the repeats inserted? From my reading of Methods it seems not.*

We thank Dr. Parker for noting that one of our major findings, that expanded structured repeat RNAs are localized to neurites, is clear. Below we address his concerns regarding the specificity of this result.

To verify the specificity of the localization of the experimental repeat RNAs, several control RNAs were examined. As suggested, we now include a figure depicting all of these constructs (Figure 1—figure supplement 2 and legend) and have added additional emphasis in the manuscript on these controls, and describe these constructs here.

The control constructs arguing for specificity include MS2 RNA and (GAA)_100_-MS2 RNA. These control RNAs contain exactly the same sequences as the (CAG)_100_-MS2, (CAG)_70_-MS2, (CAG)_40_-MS2, (CAG)_20_-MS2, (CUG)_100_-MS2, (CCUG)_100_-MS2, (GGGGCC)_48_-MS2, and (GGGGCC)_3_-MS2 RNAs, except that they either have no microsatellite repeats (MS2 RNA), or contain 100 GAA repeats ((GAA)_100_-MS2), not predicted to have stem-loop structure). All other parts of the sequence are identical. As described in the first paragraph of the Methods section, all of these experimental and control RNAs have a 5’ leader sequence that includes 6 stop codons (two in each reading frames), an EcoRI site, a BamHI site, as well as a 3’ sequence that includes FLAG-, HA-, and Myc-tags, and 12 MS2 stem loops.

The MS2 RNA and (GAA)100-MS2 RNA controls demonstrate that the leader sequence and 3’ sequence are not sufficient to confer neuritic localization. The (GAA)_100_-MS2 control also indicates that a non-structured expanded repeat sequence present in the context of the leader and 3’ sequences is not sufficient to confer neuritic localization.

In addition we used a LacZ-MS2 RNA control. This control RNA did not include the leader sequence or the FLAG-, HA-, and Myc- tags, but did include the 12 MS2 stem loop tag, thus demonstrating that the 12 MS2 stem loops are not sufficient to confer neuritic localization to an overexpressed mRNA. SeeFigure 1—figure supplement 1 and accompanying legend.

We have added Figure 1—figure supplement 2 and we have also edited the text to make the data and findings on the control RNAs clearer to readers (please see the subsection “Neuritic subcellular localization is a common property of highly structured microsatellite repeat RNAs”).

*Would any highly expressed RNA end up in neurites? Evidence that the repeats are what allows accumulation should be either provided, or if already present, clarified by more explicit description of the experiments and control RNAs examined.*

In Figure 1 we show that endogenous GGGGCC repeat RNA is present in neurites in neurons differentiated from iPSCs derived from patients that carry a GGGGCC expansion. This RNA is endogenous, hence there is no overexpression in this system. This indicates that GGGGCC repeat disease RNA does localize to neurites when expressed endogenously.

For the experiments in rat primary culture where we employed exogenous expression, as noted above, we provide several controls including MS2 only, LacZ-MS2, and MS2 tagged (GAA)_100_ RNAs. As described above, MS2 and (GAA)_100_-MS2 have the exact same flanking sequences as the experimental RNAs, but these RNAs are not localized in neurites. Likewise LacZ-MS2, expressed from the same promoter, is not localized in neurites. These findings show that high expression of any RNA is not sufficient for localization to neurites, and that there is selectivity for the repeat RNAs in question – the structured repeat RNAs tested do localize, but the unstructured control (GAA)_100_-MS2, and the MS2 controls (MS2 and LacZ-MS2) do not. This is shown in Figure 1 and in Figure 1—figure supplement 1 and clarified as indicated in the subsections “Microsatellite repeat RNAs localize to neuritic granules” and “Neuritic subcellular localization is a common property of highly structured microsatellite repeat RNAs “and legends to Figure 1 and Figure 1—figure supplement 1.

We also show in Figure 2 that there is repeat-length specificity for neuritic localization of CAG repeat RNA, such that short CAG repeat RNA that is exogenously expressed is not efficiently localized in neurites, but increased repeat number correlates with increased localization efficiency. We have also revised the manuscript to emphasize this point in the last paragraph of the subsection “Neuritic subcellular localization is a common property of highly structured microsatellite repeat RNAs”.

Also see Discussion for (GGGGCC)_48_-MS2 vs. (GGGGCC)_3_-MS2 below, under Dr. Orr’s minor point.

*A simple figure showing the compared RNAs would be helpful.*

We thank Dr. Parker for the suggestion, and to clarify the nature of the constructs, we have added Figure 1—figure supplement 2, diagramming the constructs.

*I think it would also improve the clarity and strength of this experiment to describe the cells more completely, e.g. what percentage of cells has nuclear foci? Somatic foci? Neurite foci? Does neurite foci correlate with expression level in the soma? If not, showing so would improve the rigor of the experiment.*

We have added the requested data in the text, figures, and figure legends. The percentage of neurons with nuclear foci in the total mixed rat spinal cord neuron population is 10.0 ± 4.0% s.d. (n=108 neurons). These data are in the Results, subsection “Expanded GGGGCC microsatellite repeat RNA causes neuritic branching defects”, first paragraph, and we also show an example of a neuron with nuclear foci in a new Figure 3—figure supplement 1.

The percentage of neurons with somatic RNA in the transfected population is 37.5 ± 3.7% s.d. (n=108 neurons total), see the aforementioned paragraph in the Results section. In original Figure 1 (Results section, “Neuritic subcellular localization is a common property of highly structured microsatellite repeat RNAs”, second paragraph) we showed that 21.1 ± 3.7% s.d. of neurons (n=4 cultures, 147 neurons total) contained neuritic RNA particles. In agreement with these data 52% of the neurons with somatic RNA also have neuritic RNA (19.4 ± 7.5% s.d. of the total neuron population; n=108 neurons total), see the Results subsection, “Expanded GGGGCC microsatellite repeat RNA causes neuritic branching defects”, first paragraph.

We note that nuclear foci cannot be scored simultaneously with somatic or neuritic foci due to the limitations of the MS2 system: nuclear foci are visualized by expressing NES-CP-GFP with the MS2-tagged repeat RNA whereas somatic and neuritic foci are visualized by expressing NLS-CP-GFP with the MS2-tagged repeat RNA.

*Does neurite foci correlate with expression level in the soma? If not, showing so would improve the rigor of the experiment.*

We show in Figure 1 that neuritic GGGGCC foci are present in human iPSC-derived neurons from GGGGCC expansion carriers. Thus, the repeat RNA was detected in neurites under endogenous expression of the expanded GGGGCC repeat RNA in patient iPSC-derived neurons.

To address Dr. Parker’s concern in greater detail, we have also now tested whether expression levels of the exogenous RNA correlated with neuritic localization in primary rat neurons. Expression level of neurons was quantified using ImageJ on digital immunofluorescent images, as now detailed in the Methods, subsections “Data analysis” and “Neuron culture and immunostain”. We do not find a significant correlation between neurons with neuritic (GGGGCC)_48_-MS2 RNA particles and expression level in the soma. Indeed, neurons with neuritic (GGGGCC)_48_-MS2 RNA did not have significantly higher expression level in the soma than neurons without neuritic (and only somatic) (GGGGCC)_48_-MS2 RNA (p value is >0.07). This is now indicated in the Results (subsection “Expanded GGGGCC microsatellite repeat RNA causes neuritic branching defects”), shown in a new Figure 3, and in Figure 3 legend.

*2) A second key contribution is the conclusion that neurite RNA limits arborization. To make this rigorous it would be important to again show that this only correlates with neurite RNA and not soma foci, nuclear foci, or expression level. Since one is showing correlations, being sure to examine if there are other possible correlations is important.*

We appreciate Dr. Parker’s concern and want to clarify that, in the original Figure 3, we defined neurons that had soma foci but lacked neuritic foci as “Non-Neuritic”. We have now replaced the label “Non-Neuritic” with “Somatic” in Figure 3 to avoid confusion. Neurons with somatic but not neuritic RNA have significantly more primary branches (6.1 ± 2.1 s.d.; n=20 neurons) than neurons that also contain neuritic RNA (3.8 ± 1.6 s.d.; n=12 neurons), see new Figure 3 and the Results subsection “Expanded GGGGCC microsatellite repeat RNA causes neuritic branching defects”, first paragraph.

As Dr. Parker suggests, we now add new data examining the relationships between nuclear foci and branching. We examined neurons co-expressing NES-CP-GFP and (GGGGCC)48-MS2: these data show that neurons with nuclear foci have significantly more primary branches (6.2 ± 1.3 s.d.; n=12 neurons) than neurons that contain neuritic RNA (3.8 ± 1.6 s.d.; n=12 neurons), see new Figure 3 and the aforementioned paragraph in the Results section. We also provide a tracing of a neuron with nuclear foci in new Figure 3, and a micrograph of one such neuron in new Figure 3—figure supplement 1.

Further, we add new data examining the relationship between expression level and branching, determined by scoring the fluorescence intensity of outlined neuronal cell bodies using imageJ, and scoring primary branches in neurons with a minimum of 2 primary branches, and a cell body diameter that was larger than 20 μm. New Figure 3 shows the expression level vs. the number of primary branches in neurons co-expressing NLS-CP-GFP and (GGGGCC)_48_-MS2 – we do not find a correlation between expression level and primary branch number, and in Figure 3 legend.

*3) It was unclear how the authors are concluding there are changes in the levels of mRNP granule components. As I read the work, it seems they are showing a change in average intensity in granules when the repeat RNA is expressed. Is this due to less protein in the cells? To less granule formation? This should be clarified.*

The changes in mRNP granule components were scored in iPSNs derived from human carriers of GGGGCC expansions, as compared to iPSNs derived from non-carrier controls; there was no transfection in this experiment (Figure 5).

For PSD-95 (subsection “Misregulation of transport granule components in iPSNs from GGGGCC microsatellite expansion carriers”, second paragraph) and CPEB3 (third paragraph): we examined the number of foci and the total protein levels using imaging analysis. The number of foci was scored by automatic thresholding (to avoid user bias) and particle analysis using ImageJ as described in Materials and methods, subsection “Data analysis”. Total levels of protein were scored as total pixel intensity per neuron using ImageJ; the non-carrier control pixel intensity was normalized to one.

For all of these analyses, carrier and control iPSNs were stained in the same experiment and captured with the same confocal/laser settings. For FMRP (see Results subsection “Misregulation of transport granule components in iPSNs from GGGGCC microsatellite expansion carriers”, second paragraph), total protein levels were assessed as for PSD-95 and CPEB3. Based on these data we conclude that there are more PSD-95 and CPEB3 granules and more CPEB3, FMRP and PSD-95 protein in carrier-derived iPSNs.

To make this clearer in Figure 5, we change the Y axis label from “Normalized Intensity” to “Normalized Protein Levels” and clarified these details in the Methods, subsection “Data analysis”, and in the figure legend to Figure 5.

*4) The presentation of the effects on neurite arborization is difficult to follow. I request the following alterations:*

*A) For the experiments in rat neurons it would be appropriate to examine each neuron examined for the overall level of the repeat RNA, the presence of soma foci and nuclear foci. This is an attempt to show that the correlation with neurite foci is at least as strong a correlation as one can assess.*

Please see Specific Comment 2, which addresses all of these points.

*B) For the experiments in flies I request the following:*

*i) A clear description of the differences between the "control" RNA and the (GGGGCC)_48_ RNA. Are they the same mRNAs just with an insertion of repeats? Or are they actually different? (dsRed vs. just a repeat RNA?) From my reading of Methods it seems they are different RNAs.*

The control used for the (GGGGCC)_48_ repeat expressed in da neurons is the *UAS-DsRed* transgene, previously described as a control for expanded CAG repeat constructs (Li et al., 2008). This transgene encodes DsRed expressed from the same *UAS* vector as *UAS*-(GGGGCC)_48_.

A clear description of the constructs is now provided, in the main text (subsection “Expanded GGGGCC microsatellite repeat RNA causes neuritic branching defects”, last paragraph), in the Methods section (subsection “*Drosophila* strains”), and in a new Figure 1—figure supplement 2 depicting the constructs, as well as Figure 4 and Figure 4—figure supplement 1.

*ii) For comparison of the effects of the (GGGGCC)_48_ expansion, the key comparison is the control to the (GGGGCC)_48_ expression lines at early and late time. The figures would be easier to interpret if Figure 4 was comparison of early time points for control and (GGGGCC)_48_ and Figure 4 was comparison of late control to late (GGGGCC)_48_ RNA.*

We appreciate Dr. Parker’s point that the comparison between early control to early (GGGGCC)_48_, and between late control to late (GGGGCC)_48_ are key for showing the effects of GGGGCC expansion.

A comparison between beige bars in Figure 4 shows the difference between early control and early GGGGCC expansion, whereas a comparison between pink bars in Figure 4 shows the difference between late control to late GGGGCC expansion. The axes of our graphs are aligned and identical, and the graphs are positioned next to each other to allow for this cross-panel comparison.

The original Figure 4 we find optimal both for addressing the question of whether the arborization defects reflect a lack of growth or a degeneration of pre-established neuritic branches, while still showing the effects of GGGGCC expansion and the progression of arborization from early to late stages in control and in GGGGCC expansion.

Because a direct comparison of early control to early (GGGGCC)_48_, and of late control to late (GGGGCC)_48_ in new graphs requires showing exactly the same data that is displayed in Figure 4, we have instead clarified the text in the Results section (subsection “Expanded GGGGCC microsatellite repeat RNA causes neuritic branching defects”, third paragraph), and have added new graphs displaying the data as requested by Dr. Parker to a supplemental figure (see Figure 4—figure supplement 1).

*iii) Earlier work (Lee et al., 2003) had shown that loss of function mutations in FMR result in higher-order dendritic branches, while the overexpression of FMR dramatically decreases dendritic branching. Given this it is surprising that no effects of FMR OE or RNAi are seen in the wild-type control. It might be worth commenting on this and clarifying that they see a different result. I raise this issue since OE of many FMR interacting proteins decreased dendritic branching in similar experiments (Cziko et al., 2009; Barbee et al., 2006). Perhaps a different driver is being used that is not as strong, which should just be clarified?*

*I would suggest that these key wild-type controls be included in the main text and presented in the same manner with the same type of images as the (GGGGCC)_48_ lines since this is critical to interpretation of the experiment.* As Dr. Parker points out, Lee et al. (2003) described da branching defects in *dFMR1* loss of function and overexpression studies.

In the case of loss of function, they generated *dFMR1* null animals and described a mild excess branching in ventral da neurons in segments 5 and 6 (Thoracic segments T2 and T3); they note that these changes are subtle and the da neuron branching characteristics are overlapping with the normal variation in wild type, but with a shift in distribution. They do not mention an effect on dorsal da neurons for *dFMR1* loss of function.

In our analysis, we have examined exclusively dorsal da neurons in Abdominal segments A3 and A4 (see Figure 4 legend, and Figure 4—figure supplement 1 legend) and we find that knockdown of *dFMR1* using a *UAS-dFMR1* RNAi line reveals little, if any, effect on dorsal da neuron branching in a wild type background (see Figure 4—figure supplement 1), whereas we see strong mitigation of (GGGGCC)_48_ induced branching defects in a (GGGGCC)_48_ background (Figure 4). Thus, we note that differences between our observations and those of Lee et al. likely reflect both partial rather than complete loss of *dFMR1* function, and examination of dorsal da neurons from abdominal segments rather than ventral da neurons from thoracic segments.

In their overexpression studies, Lee et al. used Gal4 line 109(2)80 to drive expression of *dFMR1*. They describe a reduction in the number of terminal dendritic processes and a decrease in length of the remaining terminal processes in both ventral and dorsal da neurons (in Thoracic segments T2 and T3). We used a different Gal4 driver line (Gal4_477_) to drive expression of *dFMR1*, and scored dorsal da neurons in Abdominal segments A3 and A4; we found no adverse effect on dorsal da neurons in a wild type background (Figure 4), whereas *dFMR1* expression in a (GGGGCC)_48_ background dramatically enhanced the expanded repeat-induced phenotype (Figure 4). Hence, regarding *dFMR1* upregulation, the differences between our observations and those of Lee et al. likely reflect the use of different driver lines and the scoring of a different set of da neurons (T2 and T3 in Lee et al., and A3, A4 here). These differences are now noted in the subsection “Transport granule components modulate GGGGCC-induced branching defects in *Drosophila* da neurons”.

We have moved these critical controls to Figure 4, as indicated above. In addition, we also move panels for control experiments for *orb2* down- and upregulation to Figure 4, respectively. The effects of *orb2* RNAi and overexpression on da neuron morphology have not been previously described.

As suggested we now also include tracings of da neurons with Gal4_477_ driven upregulation of *dFMR1* (Figure 4), upregulation of *orb2* (Figure 4), downregulation of *dFMR1* (Figure 4), and downregulation of *orb2* (Figure 4).

Reviewer #2:

*Since discovery of the C9ORF hexanucleotide repeat as a major cause of inherited ALS/FTD much effort has gone into understanding its pathogenic basis. This study presents data in support of the concept that the repeat, based on its structural features, directs repeat-containing RNA to transport granules in neurons. Interestingly, they show that the ability of the RNA to form a structure is critical for its targeting to RNA granules – for example the Frataxin repeat that does not for a structure did not localize to granules. They go onto demonstrate that association of repeat-RNA to granules is linked to neurite degeneration and alterations in proteins, whose translation is regulated by FMRP and Orb2. Overall this a strong study that uses results from several approaches and model systems to support their novel conclusion that repeat containing RNA disrupts RNA transport granule function that may be another pathogenic mechanism. A minor point that I think should be addressed is:*

*In the subsection “Neuritic subcellular localization is a common property of highly structured microsatellite repeat RNAs”, the authors describe that the number of granules with wt G4C2 RNA is much higher than those form in the presence of a G4C repeat with 48 repeats. Does this suggest that while targeting information is retained by expanded repeat the ability to form granules is compromised?*

Dr. Orr correctly notes that the fraction of arbors with expanded GGGGCC repeat RNA foci were fewer than for unexpanded GGGGCC repeat RNA, and we agree that this might indicate that the expanded repeat is assembled into granules less efficiently, or alternatively that arbors with expanded GGGGCC repeat RNA degenerate. We thank Dr. Orr for highlighting this point, and now include this speculation in the Results section (at the end of the subsection “Neuritic subcellular localization is a common property of highly structured microsatellite repeat RNAs”).